# CLIPSelf: Vision Transformer Distills Itself for Open-Vocabulary Dense Prediction

**Size Wu**[1]    **Wenwei Zhang**[1]    **Lumin Xu**[2]
**Sheng Jin**[3,4]    **Xiangtai Li**[1]    **Wentao Liu**[4,5]    **Chen Change Loy**[1]
[1] S-Lab, Nanyang Technological University    [2] The Chinese University of Hong Kong
[3] The University of Hong Kong    [4] SenseTime Research and Tetras.AI    [5] Shanghai AI Laboratory
`size001@e.ntu.edu.sg`    `ccloy@ntu.edu.sg`

## Abstract

Open-vocabulary dense prediction tasks including object detection and image segmentation have been advanced by the success of Contrastive Language-Image Pre-training (CLIP). CLIP models, particularly those incorporating vision transformers (ViTs), have exhibited remarkable generalization ability in zero-shot image classification. However, when transferring the vision-language alignment of CLIP from global image representation to local region representation for the open-vocabulary dense prediction tasks, CLIP ViTs suffer from the domain shift from full images to local image regions. In this paper, we embark on an in-depth analysis of the region-language alignment in CLIP models, which is essential for downstream open-vocabulary dense prediction tasks. Subsequently, we propose an approach named CLIPSelf, which adapts the image-level recognition ability of CLIP ViT to local image regions without needing any region-text pairs. CLIP-Self empowers ViTs to distill itself by aligning a region representation extracted from its dense feature map with the image-level representation of the corresponding image crop. With the enhanced CLIP ViTs, we achieve new state-of-the-art performance on open-vocabulary object detection, semantic segmentation, and panoptic segmentation across various benchmarks. Models and code are released at `https://github.com/wusize/CLIPSelf`.

## 1 Introduction

Dense prediction tasks, including object detection (Girshick, 2015; Ren et al., 2015) and segmentation (Hariharan et al., 2014; Kirillov et al., 2019), have been significantly advanced in the era of deep neural networks. However, traditional detection and segmentation models are trained to recognize only a fixed set of object categories. Such a design restricts the real-world applications of these models where infinite visual concepts exist. Therefore, open-vocabulary object detection (Zareian et al., 2021) and image segmentation (Ghiasi et al., 2021), which require the detection and segmentation models to recognize and localize visual concepts unseen in the training datasets, are gaining increasing attention from the community.

Recent open-vocabulary approaches (Gu et al., 2021; Xu et al., 2022; Kuo et al., 2023; Xu et al., 2023b) are typically inspired by the Contrastive Language-Image Pre-training (CLIP) (Radford et al., 2021). As depicted in Fig. 1(a), CLIP models, particularly the variants incorporating vision transformers (ViTs), have demonstrated impressive generalization capabilities in image classification tasks, achieving exceptional zero-shot performance. To enable open-vocabulary object detection and segmentation, it is crucial to transfer the vision-language alignment of CLIP models, especially the powerful ViT-based variants, from full images to local image regions.

In light of this, we conducted a preliminary experiment to evaluate the region-language alignment of CLIP's dense features and assess their competence in local object recognition. As shown in Fig. 1(b), two approaches were tested using the region boxes annotated in the COCO dataset (Lin et al., 2014). The first approach, referred to as 'Image Crop', directly inputs the image crops that enclose the regions into CLIP and utilizes the image-level features for classification. The second

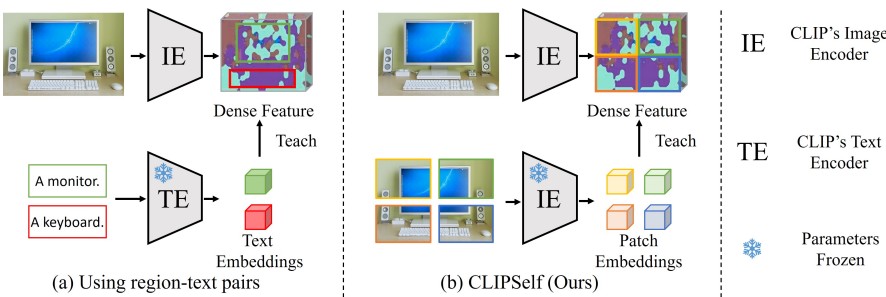

Figure 1: **(a)** CLIP ViTs exhibit excellent zero-shot ability on image classification compared with CLIP CNNs. **(b)** To classify regions, a CLIP ViT is as effective as a CLIP CNN by separately classifying the *image crop* of each region. However, it struggles when extracting region representation from the *dense feature* map for recognition. **(c)** The K-Means results of the CLIP ViT's dense feature are much noisier, demonstrating the inferiority of CLIP ViT's dense representation.

Figure 2: **(a)** Using region-text pairs to fine-tune CLIP for dense prediction tasks. These pairs are either manually annotated or generated via matching between region proposals and parsed image captions. **(b)** Our CLIPSelf does not rely on the association between text descriptions and regions, and only uses CLIP ViT's representations of image patches to learn the dense features.

approach, referred to as 'Dense Feature', first obtains the dense feature map [1] of the input image and then extracts region representations from the feature map for recognition. Our findings reveal that the dense features of the Convolutional Neural Network (CNN) based model (RN50x64) exhibit high proficiency in region classification, surpassing the approach that uses image crops. This suggests the potential of directly applying CNN-based CLIP models to open-vocabulary dense prediction tasks. In contrast, the ViT-based model (ViT-L/14) struggles with region recognition when using its dense features, despite achieving satisfactory accuracy when utilizing image-level representations of the corresponding image crops. Furthermore, the K-Means visualization of the feature maps in Fig. 1(c) provides qualitative evidence of the inferior dense representation of ViT-based CLIP models. Compared to CNN models, CLIP ViT lacks local inductive bias, hindering the smooth transfer from representing pixels of a whole image to representing pixels of a local image region. Indeed, each spot on the dense feature map of the CLIP ViT tends to encode the global image, which is not desired for local region recognition. Relevant analyses are provided in Sec. A.1. Meanwhile, notable progress has been made in building open-vocabulary object detectors based on frozen CLIP CNNs (Kuo et al., 2023; Wu et al., 2023c; Xu et al., 2023c), demonstrating competitive performance. However, applying CLIP ViTs to dense prediction tasks has proven challenging and less straightforward (Zhou et al., 2022a; Ding et al., 2023; Xu et al., 2023b). Therefore, we aim to develop an effective and general solution to address the limitations of CLIP ViTs' dense representation.

One intuitive approach to enhance CLIP ViTs for dense prediction tasks involves fine-tuning CLIP using region-text pairs as illustrated in Fig. 2(a), which establishes a direct alignment between region and language representations. However, annotating sufficient region-text pairs for training a robust region representation is resource-intensive. To address this challenge, RegionCLIP (Zhong et al., 2022) has explored the generation of pseudo labels by matching object nouns extracted from image captions with region proposals, thereby forming region-text pairs. While eliminating the need for exhaustive annotation, this approach suffers from the noisy matching between regions and object nouns. Compared with the probably imprecise text description (object noun) of a region, the image-

---

[1]The dense feature map of a CLIP ViT is obtained following Zhou et al. (2022a), which modifies the output layer of ViT for better pixel-language alignment.

level representation of the image crop enclosing the region could serve as a more reliable teacher to guide the enhancement of the region representation in the context of CLIP ViTs as shown in Fig. 1.

In this paper, we present **CLIPSelf**, a self-distillation approach that obviates the necessity of paired data associating text descriptions with image regions. Fig. 2(b) provides an overview of how CLIPSelf derives a representation conducive to dense prediction through self-distillation. Specifically, CLIPSelf fine-tunes CLIP ViTs by maximizing the cosine similarities between the region representations pooled from the dense feature map and the image representations of the corresponding image crops. The regions in our method can be obtained by partitioning an image into a grid of $m \times n$ patches. The unique design of CLIPSelf eliminates the need for a complex region-text matching process or the acquisition of additional labeled region-text pairs while effectively bridging the gap between dense and image representations of the CLIP ViTs for region recognition. Consequently, CLIPSelf significantly strengthens the vision-language alignment of the CLIP ViTs' dense features, benefiting their application to downstream open-vocabulary dense prediction tasks.

The effectiveness of CLIPSelf is validated on open-vocabulary object detection and image segmentation benchmarks. For open-vocabulary object detection, we established a two-stage baseline based on frozen CLIP ViTs, and the fine-tuned models achieved state-of-the-art performance on OV-COCO and OV-LVIS benchmarks, as well as on the transfer detection benchmark. For open-vocabulary semantic and panoptic segmentation, CLIPSelf also yields non-trivial improvements to current state-of-the-art methods, such as Cat-Seg (Cho et al., 2023) and ODISE (Xu et al., 2023a).

## 2    RELATED WORK

**Open-Vocabulary Dense Prediction.** These directions primarily include object detection (Zareian et al., 2021; Gu et al., 2021; Kuo et al., 2023) and image segmentation (Ghiasi et al., 2021; Xu et al., 2023a; 2022), aiming to recognize local visual concepts of arbitrary categories described by texts. The impressive vision-language alignment brought by Contrastive Image-Language Pre-training (CLIP) (Radford et al., 2021) has inspired numerous studies to explore the downstream application of CLIP features to these tasks. For open-vocabulary detection, a series of works (Gu et al., 2021; Wu et al., 2023a) distill knowledge from the CLIP models to recognize novel object concepts. There are also works (Kuo et al., 2023; Wu et al., 2023c) that directly build object detectors upon frozen CLIP CNNs. For open-vocabulary segmentation, the typical two-stage approaches (Xu et al., 2022; Ding et al., 2022; Xu et al., 2023a) first generate class-agnostic mask proposals and then classify the proposals with CLIP. Particularly, a concurrent work ZeroSeg (Chen et al., 2023) distills CLIP's representation on image patches to the semantic segmentation model, resembling CLIPSelf in the labelling-free nature. However, we would like to categorize ZeroSeg into downstream applications of CLIP that transfer the knowledge of CLIP to specific dense prediction models. In contrast, CLIPSelf, which facilitates the transfer of knowledge from image to local regions within the CLIP ViTs in a self-distillation manner, is posited between the upstream image-text pre-training and the downstream applications, and generally applicable to various downstream tasks.

**Vision-Language Alignment for Images and Regions.** Vision-language pre-training has given rise to models with aligned image and text representations (Frome et al., 2013; Jayaraman & Grauman, 2014; Jia et al., 2021; Kim et al., 2021; Radford et al., 2021). Recent studies on contrastive vision-language pre-training (Jia et al., 2021; Radford et al., 2021; Zhai et al., 2022) have significantly improved the generalization ability of recognition models. In particular, CLIP models (Radford et al., 2021) that are pre-trained on billion-scale image-text pairs have exhibited impressive zero-shot classification performance on a wide range of datasets. Motivated by the success in aligning image and text representations, many studies (Zhong et al., 2022; Li et al., 2022a; Liu et al., 2023; Zhang et al., 2022b; Mukhoti et al., 2023; Wu et al., 2023b) have sought to achieve vision-language alignment at the local regions of images for dense prediction tasks. Some works learn region-text alignment using annotations in visual grounding datasets (Liu et al., 2023; Krishna et al., 2017; Plummer et al., 2015) or generating pseudo region-text pairs Zhong et al. (2022); Wu et al. (2023b). There are also weakly-supervised approaches (Gupta et al., 2020; Mukhoti et al., 2023) that indirectly align region and language representations using only image-text pairs. Different from these studies, CLIPSelf facilitates the transfer of a CLIP ViT's global vision-language alignment to local regions by self-distillation, a process that circumvents associating regions with texts.

**Vision Transformers in Open-Vocabulary Learning.** For open-vocabulary or zero-shot image recognition, vision transformer (ViT) based vision-language models have demonstrated superior capability (Radford et al., 2021; Sun et al., 2023). However, ViTs have shown inferior region-language alignment in the context of open-vocabulary dense prediction, despite their success in standard dense prediction tasks (Liu et al., 2021; Zhang et al., 2021; Li et al., 2022b; Liu et al., 2022; Zhang et al., 2022a). Recent studies (Zhou et al., 2022a; Ding et al., 2023) have attempted to improve the CLIP ViTs' vision-language alignment on the dense features by modifying the output layer of ViTs or employing masked attention. However, such attempts have yielded sub-optimal results. There are also detection-oriented Swin-based foundation models that learn region-language grounding from region-text pairs, *e.g.*, GLIP (Li et al., 2022a; Zhang et al., 2022b) and Grounding DINO (Liu et al., 2023). However, these works only treat Swin as a visual backbone and separately learn the region-language grounding on a cross-modality head, without explicitly exploiting vision-language alignment in the representation of Swin Transformers. In the more relevant open-vocabulary detection literature, a few works (Minderer et al., 2022; Kim et al., 2023b;a) have sought to craft the vision-language pre-training of ViT-based open-vocabulary detectors by scaling up image-level supervision or provoking region awareness in the architecture of ViTs. For instance, CFM-ViT (Kim et al., 2023a) adopts random feature masking and window attentions in the ViTs to improve localization ability. However, such works still fall short of explicit region-language alignment, because the vision-language pre-training is conducted only on image-text pairs. To the best of our knowledge, CLIPSelf is the first work that explicitly injects strong region-language alignment into the ViT-based vision-language models. Furthermore, an enhanced variant of CLIP ViT with local window attention has been explored to further validate the promising applicability of CLIPSelf.

## 3 METHODOLOGY

### 3.1 IMAGE REPRESENTATION V.S. DENSE REPRESENTATION

A ViT-based CLIP model comprises a series of residual attention blocks. We briefly explain how the image and dense representations are obtained from the last residual attention block.

**CLIP's Image Representation.** The input to the last residual attention block is $x = (x_0, x_1, ..., x_{h \times w})^\mathsf{T}$ representing a class embedding $x_0$ and $h \times w$ image embeddings $\{x_i | i \in \{1, 2, ..., h \times w\}$. A residual attention block $z = \text{ResAttn}(x)$ can be written as:

$$q = \text{Emb}_q(x), k = \text{Emb}_k(x), v = \text{Emb}_v(x)$$

$$y = x + \text{Proj}(\text{SoftMax}(\frac{qk}{c})v), z = y + \text{FFN}(y),$$

where $c$ is a constant, Proj represents a projection layer, Emb comprises a layer norm and a projection layer, and FFN stands for a feed-forward network. Finally, the updated class embedding is used to represent the whole image: $x_{\text{image}} = z[0]$ [2].

**CLIP's Dense Representation.** To extract the dense feature map from a CLIP ViT, we follow (Zhou et al., 2022a) to slightly modify the last residual attention block. Specifically, we keep the projection layers, layer norms, and FFNs while discarding the self-attention. The modified residual attention block $z' = \text{ModifiedResAttn}(x)$ can be written as:

$$v = \text{Emb}_v(x), y' = x + \text{Proj}(v), z' = y' + \text{FFN}(y').$$

We discard the class embedding $z'[0]$ and reshape the image embeddings $z'[1 : h \times w]$ into an $h \times w$ feature map $\mathcal{X}_{\text{dense}}$, from which we can extract representations for boxes or masks by RoIAlign or mask pooling (He et al., 2017).

**Discussion and Motivation of CLIPSelf.** Although the dense features have been used in existing works (Zhou et al., 2022a; Wu et al., 2023a; Cho et al., 2023) to extract box or mask representations for dense prediction tasks, we make the first attempt to provide an in-depth analysis of the dense representation. Specifically, we compare the recognition ability of image representation and dense representation by using them to classify the region boxes annotated in the COCO dataset (Lin et al., 2014). Given an image and a region box annotation, the dense representation of the region can be

---

[2]There is a linear output layer following the last attention module. We ignore it for brevity.

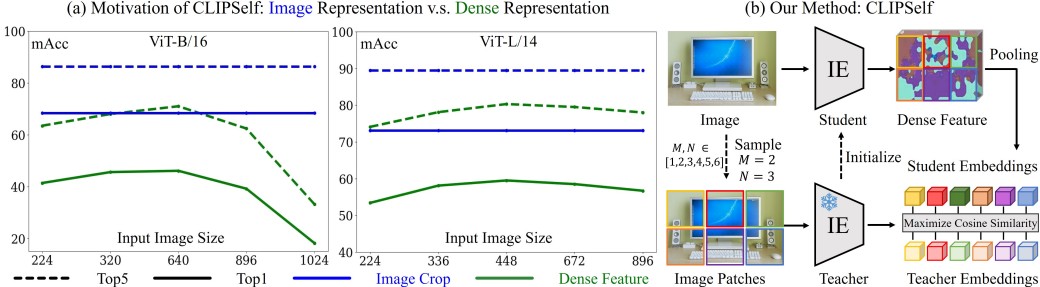

Figure 3: **(a)** Region classification using image representation (blue) and dense representation (green) of CLIP ViTs. The y-axis stands for the mean accuracy (mAcc). The x-axis is the input image size for obtaining dense feature maps (green). The input size for image representation (blue) of the image crops is fixed at $224 \times 224$ for ViT-B/16 and $336 \times 336$ for ViT-L/14. **(b)** CLIPSelf randomly splits an image into patch regions for self-distillation. Then it aligns the region representation pooled (by RoIAlign) from the dense feature map of the student to the corresponding image representation of the Teacher. **Teacher:** the original CLIP ViT; **Student:** the fine-tuned CLIP ViT.

extracted from $\mathcal{X}_{\text{dense}}$ by pooling (RoIAlign). For image representation, we crop the region box from the image first and then send it to the CLIP model to obtain $x_{\text{image}}$.

As shown in Fig. 3, the image representation (blue) outperforms the dense representation (green) on both Top1 and Top5 classification accuracy by a considerable margin across all input sizes. It is noticeable that the performance of ViTs' dense representation does not grow with the input image size, which hinders the use of the models for downstream tasks like object detection and image segmentation, where large image resolution is always desired. Please refer to Sec. A.1 for more relevant analyses and discussions on this phenomenon. Based on the results in Fig. 3(a), we believe the region representations extracted from the dense feature map can be improved by aligning them to the image representations of the corresponding image crops.

## 3.2 CLIPSELF

We propose a simple self-distillation approach, CLIPself, that allows CLIP ViTs to teach themselves by aligning the region representations pooled from dense feature maps to the image representations of the corresponding image crops as shown in Fig. 3(b). We denote the original CLIP model as *Teacher* and the fine-tuned CLIP model as *Student*. The weights of the Teacher are fixed, while the weights of the Student are initialized with the weights of the Teacher. To train the Student, we partition the image into regions and align the Student's dense representations of the regions to the corresponding image representations of the Teacher.

**Image Patches as Regions.** For self-distillation purposes, we divide an image into a grid of $m \times n$ patches. During each training iteration, $m$ and $n$ are *randomly* selected from the set $\{1, ..., M\}$ to allow different patch sizes. Empirically, we set $M = 6$ in our implementation. Experiments show that such a simple patch sampling scheme can already enhance the Student's dense representation remarkably. Moreover, the grid image patches are more effective in covering background content (referred to as 'stuff') than region proposals generated by external models, which predominantly focus on foreground objects (referred to as 'thing'). This is validated by the results of classifying stuff masks in Tab. 2.

**Self-Distillation.** Given an image, we can obtain the dense feature map $\mathcal{X}_{\text{dense}}$ from the Student as described in Sec. 3.1. Given the $m \times n$ patch regions, the Student embedding of patch $\mathcal{P}^{ij}$ ($i \in \{0, ..., m-1\}, j \in \{0, ..., n-1\}$) can be denoted as $s_{\text{dense}}^{ij} = \text{Pooling}(\mathcal{X}_{\text{dense}}, \mathcal{P}^{ij})$. Correspondingly, the teacher embedding $t_{\text{image}}^{ij}$ is obtained by sending $\mathcal{P}^{ij}$ to the Teacher for image representation. Consequently, the self-distillation loss to align student and teacher embeddings can be written as:

$$\mathcal{L} = \frac{1}{m \times n} \sum_{i=0}^{m-1} \sum_{j=0}^{n-1} (1 - \frac{s_{\text{dense}}^{ij} \cdot t_{\text{image}}^{ij}}{|s_{\text{dense}}^{ij}| \cdot |t_{\text{image}}^{ij}|}).$$

Table 1: Ablation study on the design choices of CLIPSelf. The row with blue color is our default choice in the main experiment. The experiments are on ViT-B/16 from EVA-CLIP. #1 in all the sub-tables stands for the original EVA-CLIP ViT. #5* in (c) stands for the sanity check.

<table>
<tr><td colspan="3">(a) Number of image patches</td><td colspan="3">(b) Number of learnable layers</td><td colspan="3">(c) Input image size of Student</td></tr>
<tr><td>#</td><td>Training Patch Split</td><td colspan="2" style="text-align:center">Mean Accuracy
Top1    Top5</td><td>#</td><td>Learnable Layers</td><td colspan="2" style="text-align:center">Mean Accuracy
Top1    Top5</td><td>#</td><td>Input Image Size</td><td colspan="2" style="text-align:center">Mean Accuracy
Top1    Top5</td></tr>
</table>

| # | Training Patch Split | Top1 | Top5 | # | Learnable Layers | Top1 | Top5 | # | Input Image Size | Top1 | Top5 |
|---|---|---|---|---|---|---|---|---|---|---|---|
| 1 | - | 18.2 | 33.2 | 1 | - | 18.2 | 33.2 | 1 | - | 18.2 | 33.2 |
| 2 | M=4 | 71.3 | 90.1 | 2 | 3 | 45.0 | 71.1 | 2 | 320 | 46.5 | 70.1 |
| 3 | M=6 | 72.1 | 91.3 | 3 | 6 | 59.4 | 82.3 | 3 | 640 | 67.1 | 87.7 |
| 4 | M=8 | 71.6 | 91.1 | 4 | 9 | 68.7 | 88.7 | 4 | 1024 | 72.1 | 91.3 |
| 5 | M=10 | 70.0 | 90.2 | 5 | 12 | 72.1 | 91.3 | 5* | 1024 | 52.3 | 76.4 |

## 3.3 APPLICATION TO OPEN-VOCABULARY DENSE PREDICTION

For open-vocabulary object detection, we build a two-stage detector on a frozen CLIP ViT backbone and only train the detection heads following F-VLM (Kuo et al., 2023). And we replace the backbone with the CLIPSelf fine-tuned model. As the detection task primarily targets at foreground objects, we keep using region proposals as an option for implementing CLIPSelf in the context of open-vocabulary object detection. Please refer to the appendix A.3 for the details of the detector. For semantic segmentation, our fine-tuned CLIP ViTs can serve as better initialization for the backbones of Cat-Seg (Cho et al., 2023). For panoptic segmentation, our fine-tuned ViT can be applied to the inference stage of ODISE (Xu et al., 2023a) to improve open-vocabulary ability.

## 4 EXPERIMENTS

### 4.1 ABLATION STUDY OF CLIPSELF

**Experiment Setting.** To train CLIPSelf, we use 8 A100 GPUs and set the batch size as 2 on each GPU. We train the models for 6 epochs using the AdamW (Loshchilov & Hutter, 2017) optimizer with a learning rate of $1e{-}5$ and weight decay of $0.1$. By default, we use the images in `train2017` split of COCO dataset (Lin et al., 2014), which are exactly the training images of most downstream open-vocabulary benchmarks. All experiments in Sec. 4.1 are conducted on the ViT-B/16 from EVA-CLIP (Sun et al., 2023) considering its high efficiency and capacity, and we assume using image patches for self-distillation unless otherwise stated. The mean accuracy (mAcc) of classifying region boxes annotated in COCO's `val2017` split is used as the indicator for evaluation.

**Design Choices.** We conduct ablation studies on design choices of CLIPSelf, namely *patch split*, *trainable layers*, and *input size of the Student*. For the *patch split*, an image is split into a grid of $m \times n$ patches, where $m$ and $n$ are randomly sampled from $\{1, ..., M\}$. In Tab. 1a, we show the results when $M$ is set as 4, 6, and 8. The results support our choice of $M = 6$ as it achieves the highest accuracy. For the number of *trainable layers*, we experiment with updating the last 3, 6, 9, and 12 attention layers of ViT-B/16. As shown in Tab. 1b, the region classification accuracy consistently improves with more trainable layers. Therefore, we choose to update all the attention modules in our implementation of CLIPSelf. For the *input size of the Student*, we experiment with different input image sizes ($320 \times 320, 640 \times 640, 1024 \times 1024$). The input to the Teacher is fixed at $224 \times 224$. For non-square images, we pad zero values to the bottom and right of the images after color normalization. As shown in Tab. 1c, training with larger images for the Student improves the performance of region classification. Therefore, we set $1024 \times 1024$ as the default input size for the Student. However, considering the memory cost, we do not further increase the image size. For the input size and trainable layers on ViT-L/14, please refer to the appendix A.2.

**Sanity Check on Input Image Size.** The aforementioned ablation study on input image size of the Student reveals the efficacy of training with larger inputs, which raises a question on whether the improvement of CLIPSelf is merely contributed to training the Student with higher image resolution instead of the region-wise supervision in the self-distillation. Therefore, we add a sanity check to assess to what extent the dense representation is enhanced by merely increasing the image size. Specifically, we train the Student with input size $1024 \times 1024$ using the image-level supervision from COCO Caption dataset (Chen et al., 2015). As shown in Tab. 1c(#5), image-level supervision

Table 2: Enhancement of dense representation. We report the Top1 and Top5 mean accuracy on classifying boxes and panoptic masks (thing and stuff).

| # | Model | Method | Region Proposals | Boxes Top1 | Boxes Top5 | Thing Masks Top1 | Thing Masks Top5 | Stuff Masks Top1 | Stuff Masks Top5 |
|---|-------|--------|------------------|------|------|------|------|------|------|
| 1 | ViT-B/16 | - | - | 18.2 | 33.2 | 20.6 | 36.5 | 18.4 | 43.5 |
| 2 | ViT-B/16 | CLIPSelf | ✗ | 72.1 | 91.3 | 74.4 | 91.8 | **46.8** | **80.2** |
| 3 | ViT-B/16 | CLIPSelf | ✓ | **74.0** | **92.6** | **76.3** | **92.8** | 36.8 | 75.0 |

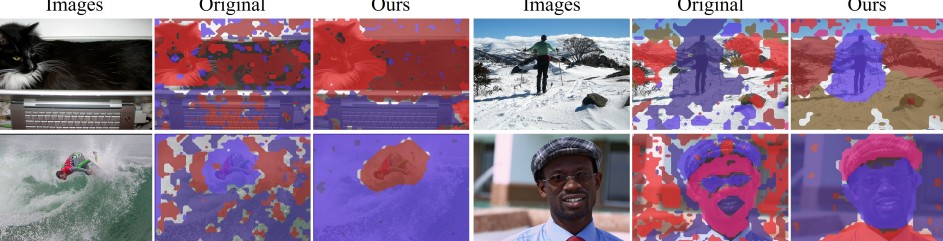

Figure 4: K-Means visualization of the dense feature maps of CLIP ViT. We show the raw images, the K-Means results of the original model, and those of our fine-tuned model by CLIPSelf.

with large input size only improves the Top1 mAcc to 52.1%, lagging behind the result of our self-distillation approach (72.1%) by a considerable margin.

## 4.2 ENHANCEMENT OF DENSE REPRESENTATION BY CLIPSELF

**Quantitative Evaluation.** We conduct a comprehensive evaluation on the enhancement of CLIP ViT's dense representation. In addition to the region box classification introduced in Sec. 4.1, we also report the mAcc of classifying panoptic masks (thing and stuff) annotated in COCO Panoptic dataset (Kirillov et al., 2019). The mask embeddings for classification are extracted from the dense feature maps of the CLIP ViT by mask pooling (He et al., 2017). As shown in Tab. 2(#1&#2), CLIP-Self not only improves the ViTs' recognition ability for region boxes but also for panoptic masks, which establishes CLIPSelf as a general solution to both open-vocabulary object detection and open-vocabulary image segmentation. Further results of various ViT variants are in the appendix A.2.

**Using Region Proposals.** We also implement CLIPSelf using region proposals generated by a region proposal network (RPN) trained on COCO's `train2017` split. To satisfy the open-vocabulary setting, the RPN is trained solely using annotations of the 48 base object categories defined in OV-COCO. As shown in Tab. 2(#3), leveraging region proposals boosts the classification accuracy for foreground instances, including object boxes and thing masks. However, this approach exhibits reduced proficiency in recognizing background contents (stuff masks) since the training of CLIPSelf has primarily emphasized foreground instances. Consequently, the utilization of region proposals is considered an alternative option only for open-vocabulary object detection.

**Qualitative Results.** We present visualizations of the CLIP ViT's dense representation by employing K-Means clustering (Lloyd, 1982) on the dense feature maps, where pixels of high cosine similarities are grouped into clusters. For clarity, we discard clusters with only a few pixels. As depicted in Fig. 4, our fine-tuned CLIP ViT demonstrates improved accuracy in gathering pixels belonging to the same object into a single cluster, *e.g.*, the 'human face' and the 'hat' in the bottom right example. Additionally, our CLIPSelf model exhibits a reduction in false positive clusters, which either cover a small portion of an object or include pixels from different objects. The improved K-Means cluster results serve as visible evidence of the enhancement of the CLIP ViT's dense representation.

## 4.3 APPLICATION TO OPEN-VOCABULARY TASKS

**Experiment Setting.** We employ the refined CLIP ViTs to open-vocabulary dense prediction tasks. To ensure a fair comparison on each open-vocabulary benchmark, we implement CLIP-Self using the training set of the corresponding benchmark. Concretely, for open-vocabulary detection on OV-COCO benchmark, the `train2017` split of COCO dataset (Lin et al., 2014) is used for self-distillation. For the OV-LVIS benchmark, we use the images from the `train` split of LVIS v1.0 (Gupta et al., 2019). For open-vocabulary image segmentation tasks, we use COCO's `train2017` split as training dataset for both the semantic and panoptic segmentation benchmarks.

Table 3: Results on open-vocabulary object detection. 'L', 'B' and 'H' in ViT-based methods stand for base, large and huge model sizes. '/16' and '/14' stand for the downsample ratio of input images.

(a) OV-COCO benchmark

| Method | Backbone | $AP_{50}^{novel}$ |
|---|---|---|
| ViLD (Gu et al., 2021) | RN50 | 27.6 |
| Detic (Zhou et al., 2022b) | RN50 | 27.8 |
| F-VLM (Kuo et al., 2023) | RN50 | 28.0 |
| OV-DETR (Zang et al., 2022) | RN50 | 29.4 |
| BARON-KD (Wu et al., 2023a) | RN50 | 34.0 |
| CORA (Wu et al., 2023c) | RN50x4 | 41.7 |
| CORA+ (Wu et al., 2023c) | RN50x4 | 43.1 |
| PromptOVD (Song & Bang, 2023) | ViT-B/16 | 30.6 |
| RO-ViT (Kim et al., 2023b) | ViT-L/16 | 33.0 |
| CFM-ViT (Kim et al., 2023a) | ViT-L/16 | 34.1 |
| F-ViT | ViT-B/16 | 17.5 |
| F-ViT+CLIPSelf | ViT-B/16 | 37.6 |
| F-ViT | ViT-L/14 | 24.7 |
| F-ViT+CLIPSelf | ViT-L/14 | **44.3** |

(b) OV-LVIS benchmark

| Method | Backbone | $mAP_r$ |
|---|---|---|
| ViLD (Gu et al., 2021) | RN50 | 16.6 |
| OV-DETR (Zang et al., 2022) | RN50 | 17.4 |
| BARON-KD (Wu et al., 2023a) | RN50 | 22.6 |
| CORA+ (Wu et al., 2023c) | RN50x4 | 28.1 |
| F-VLM (Kuo et al., 2023) | RN50x64 | 32.8 |
| PromptOVD (Song & Bang, 2023) | ViT-B/16 | 23.1 |
| OW-ViT (Minderer et al., 2022) | ViT-L/14 | 25.6 |
| RO-ViT (Kim et al., 2023b) | ViT-L/16 | 32.4 |
| CFM-ViT (Kim et al., 2023a) | ViT-L/16 | 33.9 |
| RO-ViT (Kim et al., 2023b) | ViT-H/16 | 34.1 |
| F-ViT | ViT-B/16 | 11.5 |
| F-ViT+CLIPSelf | ViT-B/16 | 25.3 |
| F-ViT | ViT-L/14 | 24.2 |
| F-ViT+CLIPSelf | ViT-L/14 | **34.9** |

Table 4: Results on open-vocabulary semantic segmentation.

| Method | Model | ADE-150 | | ADE-847 | | PASCAL Context | |
|---|---|---|---|---|---|---|---|
| | | mIoU | mAcc | mIoU | mAcc | mIoU | mAcc |
| SAN (Xu et al., 2023b) | ViT-B/16 | 27.5 | 45.6 | 10.1 | 21.1 | 53.8 | 73.0 |
| SAN (Xu et al., 2023b) | ViT-L/14 | 32.1 | 50.7 | **12.4** | 25.2 | 57.7 | 77.6 |
| Cat-Seg (Cho et al., 2023) | ViT-B/16 | 27.2 | 41.2 | 8.4 | 16.6 | 57.5 | 74.0 |
| Cat-Seg (Cho et al., 2023) | ViT-L/14 | 31.5 | 46.2 | 10.8 | 20.5 | 62.0 | 78.3 |
| Cat-Seg+CLIPSelf | ViT-B/16 | 29.0 | 46.0 | 9.3 | 20.1 | 58.0 | 75.3 |
| Cat-Seg+CLIPSelf | ViT-L/14 | **34.5** | **54.8** | **12.4** | **25.4** | **62.3** | **80.7** |

Table 5: Results on open-vocabulary panoptic segmentation. † means the results are obtained by running ODISE's officially released code and model.

| Method | Model | Score | | COCO Panoptic | | | ADE20K | | |
|---|---|---|---|---|---|---|---|---|---|
| | | CLIP | Pred | PQ | mAP | mIoU | PQ | mAP | mIoU |
| ODISE (Xu et al., 2023a)† | ViT-L/14 | ✓ | ✗ | 27.6 | 26.2 | 23.7 | 15.3 | 9.8 | 17.3 |
| ODISE (Xu et al., 2023a)† | ViT-L/14 | ✓ | ✓ | 45.3 | 38.1 | **52.3** | 22.9 | 13.4 | 28.5 |
| ODISE+CLIPSelf | ViT-L/14 | ✓ | ✗ | 35.1 | 30.9 | 36.7 | 19.5 | 10.6 | 24.5 |
| ODISE+CLIPSelf | ViT-L/14 | ✓ | ✓ | **45.7** | **38.5** | **52.3** | **23.7** | **13.6** | **30.1** |

**Open-Vocabulary Object Detection.** Following F-VLM (Kuo et al., 2023), which freezes the CLIP backbone, we introduce a two-stage detector baseline built on frozen CLIP ViTs, called *F-ViT*. To extract multi-scale feature maps from the frozen backbone, we interpolate the feature maps from the intermediate layers of ViTs. We use the ViTs from EVA-CLIP (Sun et al., 2023) for our main experiments for their efficiency and high capacity. The results are shown in Tab. 3. By replacing the CLIP ViTs refined by CLIPSelf, the performance is significantly improved (44.3 vs 24.7 $AP_{50}^{novel}$ on OV-COCO and 34.9 vs 24.2 $mAP_r$ on OV-LVIS) and our detectors achieves new state-of-the-art results on the two benchmarks. More details and experimental results can be found in Sec. A.3.

**Open-Vocabulary Semantic Segmentation.** We apply CLIPSelf fine-tuned models to Cat-Seg (Cho et al., 2023), where the dense features of CLIP ViTs (ViT-B/16 and ViT-L/14 from OpenAI) are used in a cost-aggregation module. The segmentation model is trained on COCO Stuff (Caesar et al., 2018) and evaluated on ADE20K (Zhou et al., 2017) (ADE-150 and ADE-847) and PASCAL Context (Mottaghi et al., 2014) datasets. As shown in Tab. 4, CLIPSelf leads to performance improvement across all the test datasets.

**Open-Vocabulary Panoptic Segmentation.** We first reproduce ODISE (Xu et al., 2023a) using the original CLIP model (ViT-L/14 from OpenAI) by running its officially released model and code[3]. Subsequently, we apply our fine-tuned model during the inference stage of ODISE, where the CLIP score and the score predicted by the mask generator are fused to classify the panoptic masks. The model is trained on the COCO Panoptic (Lin et al., 2014) dataset and evaluated on ADE20K (Zhou et al., 2017) dataset. Tab. 5 presents the results of using the CLIP score only and using the fused score. The observed improvements in Tab. 5 indicate the enhanced recognition capability of ViT's dense representation for recognizing panoptic masks.

---

[3]https://github.com/NVlabs/ODISE

Table 6: Comparison with using region-text pairs. Models are from EVA-CLIP (Sun et al., 2023).

| Model | Method | Region Proposals | Boxes | | Thing Masks | | Stuff Masks | | OV-COCO | |
|---|---|---|---|---|---|---|---|---|---|---|
| | | | Top1 | Top5 | Top1 | Top5 | Top1 | Top5 | $AP_{50}^{novel}$ | $AP_{50}^{base}$ |
| ViT-B/16 | Region-Text | ✓ | 71.1 | 90.7 | 73.7 | 91.4 | 34.2 | 68.6 | 34.4 | 54.2 |
| ViT-B/16 | CLIPSelf | ✓ | **74.0** | **92.6** | **76.3** | **92.8** | **36.8** | **75.0** | **37.6** | **54.9** |

Table 7: Exploring local window attention (denoted as 'WindowAttn'). The 'GlobalAttn' means the global attention used in the original CLIP ViTs. Models are from EVA-CLIP (Sun et al., 2023).

| # | Model | CLIPSelf | Attention | Boxes | | Thing Masks | | Stuff Masks | | OV-COCO | |
|---|---|---|---|---|---|---|---|---|---|---|---|
| | | | | Top1 | Top5 | Top1 | Top5 | Top1 | Top5 | $AP_{50}^{novel}$ | $AP_{50}^{base}$ |
| 1 | ViT-B/16 | ✗ | GlobalAttn | 18.2 | 33.2 | 20.6 | 36.5 | 18.4 | 43.5 | 17.5 | 41.0 |
| 2 | ViT-B/16 | ✗ | WindowAttn | 34.7 | 60.3 | 40.6 | 64.8 | 30.9 | 61.2 | 19.4 | 48.5 |
| 3 | ViT-B/16 | ✓ | GlobalAttn | 72.1 | 91.3 | 74.4 | 91.8 | 46.8 | 80.2 | 33.6 | 54.2 |
| 4 | ViT-B/16 | ✓ | WindowAttn | 73.3 | 91.7 | 74.9 | 92.1 | 48.6 | 81.0 | 33.6 | 55.9 |

Table 8: Performing self-distillation on CC3M (Sharma et al., 2018). The model (ViT-B/16 from EVA-CLIP (Sun et al., 2023)) is trained on CC3M for 1 epoch.

| Model | CLIPSelf | Boxes | | Thing Masks | | Stuff Masks | | OV-COCO | | OV-LVIS | | |
|---|---|---|---|---|---|---|---|---|---|---|---|---|
| | | Top1 | Top5 | Top1 | Top5 | Top1 | Top5 | $AP_{50}^{novel}$ | $AP_{50}^{base}$ | $mAP_r$ | $mAP_c$ | $mAP_f$ |
| ViT-B/16 | ✗ | 18.2 | 33.2 | 20.6 | 36.5 | 18.4 | 43.5 | 17.5 | 41.0 | 11.5 | 12.3 | 20.6 |
| ViT-B/16 | ✓ | **72.1** | **91.5** | **74.5** | **92.0** | **49.5** | **81.6** | **35.8** | **54.6** | **26.6** | **21.7** | **29.2** |

## 4.4 DISCUSSION

**Comparison with Using Region-Text Pairs.** To demonstrate the advantage of CLIPSelf, we compare it with the method using region-text pairs. We follow the principle of RegionCLIP (Zhong et al., 2022) that matches region proposals with object nouns to generate region-text pairs to implement the compared method. For a fair comparison, CLIPSelf also employs region proposals. More details on the compared method are in Sec. A.4. As indicated in Tab. 6 (#1 and #2), CLIPSelf outperforms the method using noisy region-text pairs by a large margin.

**Beyond Plain ViTs.** To verify CLIPSelf's applicability to different models, we exploit architectures beyond plain ViTs (global attention). Unfortunately, there are no public CLIP-like ViTs equipped with local attentions, *e.g.* Swin Transformer (Liu et al., 2021). Relevant methods like GLIP (Li et al., 2022a) are built under different benchmarks and simply treat Swin as the backbone without explicitly exploring its region-language alignment. Therefore, we opt for developing local window attention based ViTs from the current CLIP ViTs. Specifically, we replace the original global attention with $4 \times 4$ window attentions. For the global image token (class token), we replicate it when splitting the image into windows and average the class token of each window when merging the windows. This modification improves region recognition and open-vocabulary detection (#1 and #2), but lags largely behind using CLIPSelf for self-distillation (#2 and #3). Importantly, CLIPSelf can consistently improve the CLIP model with window attention (#2 and #4). Therefore, we believe the effectiveness of CLIPSelf can be extrapolated to a wider range of model architectures.

**Training on CC3M.** The experiments above mainly use the downstream COCO dataset for self-distillation to ensure fair comparison with prior methods, especially distillation-based and frozen CLIP-based approaches. We also consider conducting self-distillation on the out-of-domain CC3M dataset (Sharma et al., 2018). As shown in Tab. 8, the model fine-tuned on CC3M exhibits consistent improvement on zero-shot box and mask recognition as well as open-vocabulary object detection.

## 5 CONCLUSION

In this paper, we undertake a comprehensive analysis of the dense representation of ViT-based CLIP models. Based on this analysis, we introduce CLIPSelf, which fine-tunes CLIP ViTs' dense representation via self-distillation without relying on region-text pairs. The fine-tuned CLIP ViTs significantly surpass the performance of the original models when applied to open-vocabulary object detection and image segmentation. Furthermore, we validate CLIPSelf's promising applicability by exploring local window attentions beyond plain ViTs and implementing CLIPSelf on web data (CC3M). In conclusion, our CLIPSelf serves as a straightforward yet effective solution to enhance the dense representation of CLIP ViTs, which is critical to open-vocabulary dense prediction tasks.

## 6 ACKNOWLEDGEMENTS

This research is supported by the National Research Foundation, Singapore under its AI Singapore Programme (AISG Award No: AISG3-PhD-2023-08-048T), the RIE2020 Industry Alignment Fund – Industry Collaboration Projects (IAF-ICP) Funding Initiative, as well as cash and in-kind contribution from the industry partner(s).

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

# A APPENDIX

## A.1 CLIP MODELS' DENSE REPRESENTATION

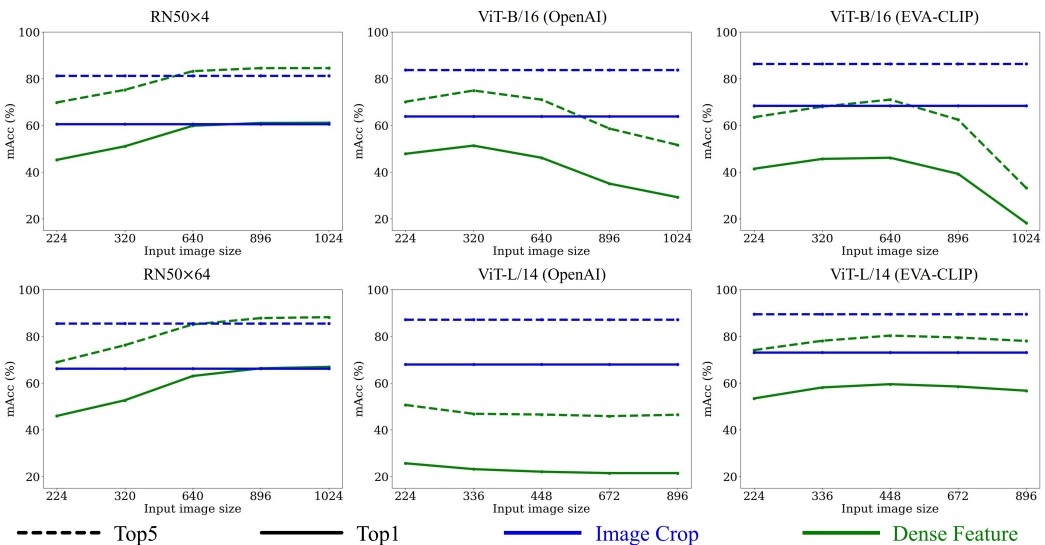

Figure A1: Region classification using CLIP Models. The x-axis of the figures stands for the input size to obtain dense features. The input size for the image-level representation of the image crops is fixed at $288 \times 288$ for RN50×4, $448 \times 448$ for RN50×64, $224 \times 224$ for ViT-B/16 and $336 \times 336$ for ViT-L/14.

Table A1: Retrieval experiment on images and regions. The images and regions are obtained from COCO's `val2017` split.

| Model | CLIPSelf | Image Retrieval | | | Region Retrieval | | |
|---|---|---|---|---|---|---|---|
| | | R@1 | R@5 | R@10 | R@1 | R@5 | R@10 |
| ViT-B/16 | ✗ | 67.0 | 85.6 | 91.4 | 34.7 | 60.5 | 71.4 |
| ViT-B/16 | ✓ | 28.4 | 50.3 | 62.5 | 57.1 | 80.1 | 85.8 |
| ViT-L/14 | ✗ | 46.1 | 63.6 | 71.4 | 45.2 | 66.6 | 74.6 |
| ViT-L/14 | ✓ | 37.8 | 25.5 | 61.1 | 49.4 | 69.8 | 76.0 |

**Region Recognition.** We conduct a comprehensive analysis of the dense representation of CLIP models in the context of region recognition. Specifically, we evaluate the Top1 and Top5 mean accuracy (mAcc) of region classification using the dense features of CLIP models with varying input image sizes. Fig. A1 illustrates the results obtained. For CNN-based models, we observe that their dense features are highly effective for region recognition. In fact, the performance of region recognition using dense feature maps surpasses that of utilizing image-level representations of the corresponding image crops, particularly when the input image size exceeds $640 \times 640$. The remarkable alignment between regions and language exhibited by CLIP CNNs enables their straightforward application to downstream open-vocabulary dense prediction tasks. However, the trend is opposite for CLIP Vision Transformers (ViTs). The dense features of ViTs exhibit inferior performance in region classification across all input image sizes. Moreover, the accuracy tends to decrease as the input image size is further increased. Meanwhile, employing ViTs' image-level representation of the regions' image crops yields satisfactory results, indicating that it can serve as a reliable teacher for refining the dense representation. These findings shed light on the contrasting behaviors of CLIP CNNs and ViTs, emphasizing the potential benefits of leveraging image-level representations to refine the dense representation of ViTs.

**Are Dense Features Encoding Global Images?** We conduct a retrieval experiment to answer this question. Specifically, we first encode images and regions annotated in COCO' validation split. The region embeddings are obtained by cropping the region boxes and sending to the CLIP model. Then we extract dense feature maps of the images and let each location in the feature map retrieve the

Table A2: Different extraction methods to obtain dense features.

| # | Model | Source | Method | Boxes | | Thing Masks | | Stuff Masks | |
|---|---|---|---|---|---|---|---|---|---|
| | | | | Top1 | Top5 | Top1 | Top5 | Top1 | Top5 |
| 1 | ViT-B/16 | OpenAI | Image Crops | 63.8 | 83.7 | - | - | - | - |
| 2 | ViT-B/16 | OpenAI | (Zhou et al., 2022a) | 29.3 | 51.6 | 33.5 | 56.0 | 25.9 | 50.9 |
| 3 | ViT-B/16 | OpenAI | Masked Attention | 28.2 | 46.7 | 29.6 | 47.5 | 32.2 | 61.6 |
| 4 | ViT-B/16 | OpenAI | CLIPSelf | 67.4 | 88.4 | 69.4 | 88.5 | 43.4 | 76.9 |

Table A3: The enhanced dense representations for recognizing boxes, thing masks, and stuff masks.

| Model | Source | CLIPSelf | Boxes | | Thing Masks | | Stuff Masks | |
|---|---|---|---|---|---|---|---|---|
| | | | Top1 | Top5 | Top1 | Top5 | Top1 | Top5 |
| ViT-B/16 | OpenAI | ✗ | 29.3 | 51.6 | 33.5 | 56.0 | 25.9 | 50.9 |
| ViT-B/16 | OpenAI | ✓ | 67.4 | 88.4 | 69.4 | 88.5 | 43.4 | 76.9 |
| ViT-B/16 | EVA-CLIP | ✗ | 18.2 | 33.2 | 20.6 | 36.5 | 18.4 | 43.5 |
| ViT-B/16 | EVA-CLIP | ✓ | 72.1 | 91.3 | 74.4 | 91.8 | 46.8 | 80.2 |
| ViT-L/14 | OpenAI | ✗ | 21.4 | 45.9 | 28.3 | 52.0 | 11.8 | 27.9 |
| ViT-L/14 | OpenAI | ✓ | 68.9 | 89.6 | 70.0 | 88.0 | 35.5 | 71.5 |
| ViT-L/14 | EVA-CLIP | ✗ | 56.7 | 78.0 | 59.0 | 79.8 | 20.8 | 41.9 |
| ViT-L/14 | EVA-CLIP | ✓ | 77.1 | 93.3 | 78.7 | 93.7 | 44.4 | 78.3 |

image and region it belongs by calculating cosine similarity between the location feature and the encoded image or region features. For each retrieval, we provide 50 samples (1 positive and 49 negative samples) and calculate recall at 1, 5, and 10, respectively. As shown in Tab. A1(#1), the dense features are well matched with the corresponding images, indicating that each location on the dense feature map tends to encode a global image representation. This observation coincides with the K-Means visualization in Fig. 1 and Fig. 4 where the clustering results of the original CLIP ViT are quite diffused.

**Masked Attention.** We follow the practice in (Zhou et al., 2022a) to extract dense feature maps of CLIP ViTs. One might question whether applying a masked-attention operation could be a better cure for the inferior dense-level representation. However, as presented in Tab. A2(#3), our experiments indicate that such an approach improves the recognition of stuff masks but worsens the classification of object boxes and thing masks. Furthermore, the accuracy improvement for stuff masks achieved through the masked-attention operation lags significantly behind the performance of CLIPSelf (#4).

## A.2 CLIPSELF

**Input Image Sizes & Trainable Layers.** For the Student model, which extracts dense representations, we set the image size of input images as $1024 \times 1024$ for ViT-B/16 and $896 \times 896$ for ViT-L/14. As for the Teacher model, which takes cropped image patches as input, we set the input size as $224 \times 224$ for ViT-B/16 and $336 \times 336$ for ViT-L/14. In the training of CLIPSelf, we update the 12 attention layers of ViT-B/16 and the 24 attention layers of ViT-L/14.

**Enhancement of Dense Representation.** To summarize the improvements achieved by CLIPSelf in the context of classifying boxes and masks, we provide a summary in Tab. A3. The results clearly demonstrate that CLIPSelf effectively enhances the dense representation of all variants of ViTs.

**From Images to Regions.** As shown in the retrieval experiment in Tab. A1(#2), the models refined by CLIPSelf yield dense features that better match the corresponding regions instead of images, revealing the transfer of knowledge from global image representation to local region representation.

## A.3 OPEN-VOCABULARY OBJECT DETECTION

**Benchmark Details.** The open-vocabulary COCO (OV-COCO) benchmark, proposed in OV-RCNN (Zareian et al., 2021), splits 65 object categories in COCO dataset (Lin et al., 2014) into 48 base categories and 17 novel categories. The open-vocabulary LVIS (OV-LVIS) benchmark, proposed in ViLD (Gu et al., 2021), sets the 337 rare categories in LVIS v1.0 (Gupta et al., 2019) dataset

Table A4: Design choices of the open-vocabulary object detector. The 'Backbone LR' represents the scalar multiplied by the base learning rate (1e-4) in training. 'Inter' means using outputs of ViTs' intermediate layers.

| # | Model | FPN | Layers | Backbone LR | $AP_{50}^{novel}$ | $AP_{50}^{base}$ | $AP_{50}$ |
|---|-------|-----|--------|-------------|-------------------|------------------|-----------|
| 1 | ViT-B/16 | SimpleFPN | Last | ×0.0 | 23.1 | 46.6 | 40.4 |
| 2 | ViT-B/16 | Standard | Last | ×0.0 | 26.5 | 47.0 | 41.6 |
| 3 | ViT-B/16 | Standard | Inter | ×0.0 | **33.6** | 54.2 | 48.8 |
| 4 | ViT-B/16 | Standard | Inter | ×0.01 | 31.1 | 56.6 | 49.9 |
| 5 | ViT-B/16 | Standard | Inter | ×0.1 | 21.8 | 58.6 | 48.8 |

Table A5: Results of OpenAI models on OV-COCO and OV-LVIS benchmarks.

| Method | Model | OV-COCO | | | OV-LVIS | | | |
|--------|-------|---------|---|---|---------|---|---|---|
| | | $AP_{50}^{novel}$ | $AP_{50}^{base}$ | $AP_{50}$ | $mAP_r$ | $mAP_c$ | $mAP_f$ | $mAP$ |
| F-ViT | ViT-B/16 | 16.0 | 36.9 | 31.4 | 8.3 | 11.4 | 19.7 | 14.1 |
| F-ViT+CLIPSelf | ViT-B/16 | 29.8 | 46.9 | 42.5 | 21.6 | 16.2 | 23.8 | 20.1 |
| F-ViT | ViT-L/14 | 9.2 | 44.3 | 35.2 | 10.7 | 19.6 | 26.3 | 20.7 |
| F-ViT+CLIPSelf | ViT-L/14 | 31.3 | 49.2 | 44.6 | 24.4 | 21.1 | 27.5 | 24.2 |

Table A6: Detailed comparison on OV-COCO benchmark. † means using learnable category prompts for region classification. * stands for methods that obtain open-vocabulary ability from both CLIP visual model and additional image annotations (*e.g.*, image captions).

| Method | Backbone | $AP_{50}^{novel}$ | $AP_{50}^{base}$ | $AP_{50}$ |
|--------|----------|-------------------|------------------|-----------|
| OV-RCNN (Zareian et al., 2021) | RN50 | 17.5 | 41.0 | 34.9 |
| RegionCLIP (Zhong et al., 2022) | RN50 | 26.8 | 54.8 | 47.5 |
| RegionCLIP (Zhong et al., 2022) | RN50 | 31.4 | 57.1 | 50.4 |
| RegionCLIP (Zhong et al., 2022) | RN50x4 | 39.3 | 61.6 | 55.7 |
| ViLD (Gu et al., 2021) | RN50 | 27.6 | 59.5 | 51.2 |
| OV-DETR (Zang et al., 2022) | RN50 | 29.4 | 61.0 | 52.7 |
| PB-OVD (Gao et al., 2022) | RN50 | 30.8 | 46.1 | 42.1 |
| Detic (Zhou et al., 2022b) | RN50 | 27.8 | 51.1 | 45.0 |
| OC-OVD (Rasheed et al., 2022)* | RN50 | 36.6 | 54.0 | 49.4 |
| VLDet (Lin et al., 2023) | RN50 | 32.0 | 50.6 | 45.8 |
| F-VLM (Kuo et al., 2023) | RN50 | 28.0 | - | 39.6 |
| BARON-Cap (Wu et al., 2023a) | RN50 | 33.1 | 54.8 | 49.1 |
| BARON-KD (Wu et al., 2023a) | RN50 | 34.0 | 60.4 | 53.5 |
| BARON-Cap&KD (Wu et al., 2023a)* | RN50 | 42.7 | 54.9 | 51.7 |
| OADP (Wang et al., 2023) | RN50 | 35.6 | 55.8 | 50.5 |
| CORA (Wu et al., 2023c)† | RN50 | 35.1 | 35.5 | 35.4 |
| CORA (Wu et al., 2023c)† | RN50x4 | 41.7 | 44.5 | 43.8 |
| CORA+ (Wu et al., 2023c)†* | RN50x4 | 43.1 | 60.9 | 56.2 |
| RO-ViT (Kim et al., 2023b) | ViT-B/16 | 30.2 | - | 41.5 |
| RO-ViT (Kim et al., 2023b) | ViT-L/16 | 33.0 | - | 47.7 |
| CFM-ViT (Kim et al., 2023a) | ViT-L/16 | 34.1 | - | 46.0 |
| F-ViT | ViT-B/16 | 17.5 | 41.0 | 34.9 |
| F-ViT+Region-Text | ViT-B/16 | 34.4 | 54.2 | 49.0 |
| F-ViT+CLIPSelf (Image Patch) | ViT-B/16 | 33.6 | 54.2 | 48.8 |
| F-ViT+CLIPSelf (Region Proposal) | ViT-B/16 | 37.6 | 54.9 | 50.4 |
| F-ViT | ViT-L/14 | 24.7 | 53.6 | 46.0 |
| F-ViT+Region-Text | ViT-L/14 | 38.7 | 59.6 | 54.1 |
| F-ViT+CLIPSelf (Image Patch) | ViT-L/14 | 38.4 | 60.6 | 54.8 |
| F-ViT+CLIPSelf (Region Proposal) | ViT-L/14 | **44.3** | 64.1 | 59.0 |

as novel categories. For evaluation, we follow previous works to use box AP at IoU 0.50 on novel categories ($AP_{50}^{novel}$) as the main metric on OV-COCO, and the mean mask AP on rare categories ($mAP_r$) as the main metric on OV-LVIS.

**Implementation Details.** To train the detector, we set the batch size to 8 on each GPU and employ the AdamW optimizer with a learning rate of $1e-4$ and weight decay of $0.1$. We train the models for 3 epochs on the OV-COCO benchmark and 48 epochs on the OV-LVIS benchmark. In the main

Table A7: Detailed comparison on OV-LVIS benchmark. † means using a learnable category prompt for region classification. * stands for methods that obtain open-vocabulary ability from both CLIP visual model and additional image annotations (*e.g.*, image captions).

| Method | Backbone | $mAP_r$ | $mAP_c$ | $mAP_f$ | mAP |
|---|---|---|---|---|---|
| RegionCLIP (Zhong et al., 2022) | RN50 | 17.1 | 27.4 | 34.0 | 28.2 |
| RegionCLIP (Zhong et al., 2022) | RN50x4 | 22.0 | 32.1 | 36.9 | 32.3 |
| Detic (Zhou et al., 2022b) | RN50 | 24.9 | - | - | 32.4 |
| Detic (Zhou et al., 2022b) | SwinB | 33.8 | - | - | 47.0 |
| VLDet (Lin et al., 2023) | RN50 | 21.7 | 29.8 | 34.3 | 30.1 |
| VLDet (Lin et al., 2023) | SwinB | 26.3 | 39.4 | 41.9 | 38.1 |
| ViLD (Gu et al., 2021) | RN50 | 16.6 | 24.6 | 30.3 | 25.5 |
| OV-DETR (Zang et al., 2022) | RN50 | 17.4 | 25.0 | 32.5 | 26.6 |
| DetPro (Du et al., 2022)† | RN50 | 19.8 | 25.6 | 28.9 | 25.9 |
| BARON-KD (Wu et al., 2023a)† | RN50 | 22.6 | 27.6 | 29.8 | 27.6 |
| OADP (Wang et al., 2023) | RN50 | 21.7 | 26.3 | 29.0 | 26.6 |
| OC-OVD (Rasheed et al., 2022)* | RN50 | 21.1 | 25.0 | 29.1 | 25.9 |
| F-VLM (Kuo et al., 2023) | RN50 | 18.6 | - | - | 24.2 |
| F-VLM (Kuo et al., 2023) | RN50x4 | 26.3 | - | - | 28.5 |
| F-VLM (Kuo et al., 2023) | RN50x16 | 30.4 | - | - | 32.1 |
| F-VLM (Kuo et al., 2023) | RN50x64 | 32.8 | - | - | 34.9 |
| CORA (Wu et al., 2023c)† | RN50x4 | 22.2 | - | - | - |
| CORA+ (Wu et al., 2023c)†* | RN50x4 | 28.1 | - | - | - |
| OWL-ViT (Kim et al., 2023b) | ViT-L/14 | 25.6 | - | - | 34.7 |
| RO-ViT (Kim et al., 2023b) | ViT-B/16 | 28.0 | - | - | 30.2 |
| RO-ViT (Kim et al., 2023b) | ViT-L/16 | 32.1 | - | - | 34.0 |
| RO-ViT (Kim et al., 2023b) | ViT-H/16 | 34.1 | - | - | 35.1 |
| CFM-ViT (Kim et al., 2023a) | ViT-L/16 | 33.9 | - | - | 36.6 |
| F-ViT | ViT-B/16 | 11.5 | 12.3 | 20.6 | 15.4 |
| F-ViT+CLIPSelf (Image Patch) | ViT-B/16 | 25.3 | 21.8 | 29.1 | 25.2 |
| F-ViT | ViT-L/14 | 24.2 | 27.9 | 31.5 | 28.7 |
| F-ViT+CLIPSelf (Image Patch) | ViT-L/14 | **34.9** | 34.6 | 35.6 | 35.1 |

experiments using ViTs from EVA-CLIP (Sun et al., 2023), the input image size is set as $896 \times 896$ for ViT-L/14. For ViT-B/16, the input image size is $640 \times 640$ on OV-COCO and $1024 \times 1024$ on OV-LVIS. For potential non-square inputs, we pad zero values to the bottom and right of the images after color normalization.

**F-ViT Architecture.** In this section, we examine the design choices of the baseline open-vocabulary detector, F-ViT. To build an object detector based on ViTs, a popular approach is ViTDet (Li et al., 2022b), which utilizes a simple Feature Pyramid Network (FPN) and only employs feature maps from the last attention layer. The design of ViTDet is based on two observations: (1) the high capacity of ViT allows it to learn detection-related features without additional parameters except a standard FPN, and (2) the output of the last attention layer is trained to provide task-specific representations for object detection. However, in the course of adapting CLIP ViTs for open-vocabulary object detection where we would like to freeze the backbone to retain the vision-language alignment of CLIP (Kuo et al., 2023), these two presuppositions no longer hold since we do not update the ViT to learn the task-specific features. As shown in Tab. A4, using the ViTDet-like architecture (#1) yields the worst results. Transitioning to a standard FPN (#2) slightly improves the performance. Then, we utilize the feature maps from multiple intermediate layers of the ViT (#3) instead of solely relying on the output of the last layer. This design significantly enhances performance for both novel and base categories. Specifically, we interpolate the feature maps from layers $[3, 5, 7, 11]$ of ViT-B/16 with relative scales $\left[\frac{1}{4}, \frac{1}{8}, \frac{1}{16}, \frac{1}{32}\right]$ to the input image size. For ViT-L/14, we interpolate the feature maps from layers $[6, 10, 14, 23]$ with relative scales $\left[\frac{1}{3.5}, \frac{1}{7}, \frac{1}{14}, \frac{1}{28}\right]$ to the input image size.

**Unfreeze the Backbone.** We explore the possibility of updating the parameters of the ViT backbone in our approach. Tab. A4 (#4&#5) presents the impact on performance when gradually increasing the learning rate of the backbone. Notably, while the AP50 for base categories improves, the performance for novel categories decreases. Based on these observations, we opt to maintain a fixed ViT backbone, as the frozen architecture already achieves satisfactory results on base categories.

Table A8: Transfer evaluation of the LVIS-trained detector on PACAL VOC, COCO and Objects365.

| Method | Pascal VOC | | COCO | | | Objects365 | | |
|---|---|---|---|---|---|---|---|---|
| | $AP_{50}$ | $AP_{75}$ | AP | $AP_{50}$ | $AP_{75}$ | AP | $AP_{50}$ | $AP_{75}$ |
| ViLD (Gu et al., 2021) | - | - | 36.6 | 55.6 | 39.8 | 11.8 | 18.2 | 12.6 |
| DetPro (Du et al., 2022) | 74.6 | 57.9 | 34.9 | 53.8 | 37.4 | 12.1 | 18.8 | 12.9 |
| BARON-KD (Wu et al., 2023a) | 76.0 | 58.2 | 36.2 | 55.7 | 39.1 | 13.6 | 21.0 | 14.5 |
| RO-ViT (Kim et al., 2023b) | - | - | - | - | - | 17.1 | 26.9 | 19.5 |
| F-VLM (Kuo et al., 2023) | - | - | 39.8 | 61.6 | 43.8 | 17.7 | 27.4 | 19.1 |
| CFM-ViT (Kim et al., 2023a) | - | - | - | - | - | 18.7 | 28.9 | 20.3 |
| F-ViT+CLIPSelf | **77.6** | **59.8** | **40.5** | **63.8** | **44.3** | **19.5** | **31.3** | **20.7** |

**OVD results with OpenAI models.** In the main experiments of open-vocabulary detection, we employ ViTs from EVA-CLIP (Sun et al., 2023), given their favorable performance and high efficiency. However, it is worth noting that CLIPSelf can also be applied for other ViT variants. We report the results using ViTs from OpenAI (Radford et al., 2021) in Tab. A5. The input size is $640 \times 640$ for ViT-B/16 and $672 \times 672$ for ViT-L/14. We only use image patches for self-distillation. All the other settings remain the same as the experiments on EVA-CLIP ViTs.

**Comprehensive System-Level Comparison.** To provide a thorough and comprehensive system-level comparison of open-vocabulary object detection methods, we present detailed results for both the OV-COCO and OV-LVIS benchmarks in Tab. A6 and Tab. A7, respectively.

**Transfer Evaluation.** We evaluate the detector (ViT-L-/14) trained on OV-LVIS on the validation split of PASCAL VOC (Everingham et al., 2010), COCO (Lin et al., 2014) and Objects365 v1 (Shao et al., 2019) datasets. Our approach consistently outperforms all previous methods in the transfer evaluation.

## A.4 USING REGION-TEXT PAIRS

We adopt the methodology proposed in RegionCLIP (Zhong et al., 2022) to fine-tune CLIP ViTs using pseudo-labelled region-text pairs. Specifically, we parse object nouns from the COCO Caption dataset (Chen et al., 2015) and generate region proposals using an RPN trained on COCO's 48 base categories. To establish correspondence between object nouns and region proposals, the original RegionCLIP extracts region features from the dense feature maps of CLIP CNNs. Differently, we use the image-level features of the image crops enclosing the regions, considering the inferior dense representation of the original CLIP ViT. For a fair comparison, we also fine-tune the model on the `train2017` split of COCO dataset (Lin et al., 2014) for the same 6 epochs.

## B BROADER IMPACT

Our work contributes to an in-depth analysis of the dense representation of ViT-based CLIP models and introduces CLIPSelf, an effective approach to unleash the power of CLIP ViTs for open-vocabulary dense prediction. We are the first to build open-vocabulary object detectors upon frozen CLIP ViT backbones and achieve state-of-the-art performance. As transformers are becoming increasingly popular as a unified architecture for both vision and language tasks, it is of great significance to successfully adapt the generalization ability of CLIP ViTs from image classification tasks to dense prediction tasks. Experiments also validate the effectiveness of our CLIPSelf beyond plain ViTs, demonstrating promising applicability of CLIPSelf on a wider range of model architectures.

## C LIMITATION

Our work is built upon the pre-trained CLIP models. Therefore, the performance of CLIPSelf is greatly influenced by the visual-language alignment in the original CLIP models. How to further enhance both image and dense representation of pre-trained vision-language models will be an interesting research topic. Moreover, as there are no public CLIP-like vision-language models with the Swin Transformer (Liu et al., 2021) backbone, we are not able to directly validate the effectiveness of CLIPSelf on the Swin-based models. Recently, there are some detection-oriented Swin-based foundation models that learn region-language grounding from region-text pairs, *e.g.*, GLIP (Li et al.,

2022a; Zhang et al., 2022b) and Grounding DINO (Liu et al., 2023). We envision that CLIPSelf has the potential to further enhance these works. The self-distillation can be a strong supplement or replacement of the pseudo labelling process in these methods. However, due to resource limitation and lack of further study on the vision and language representations within these architectures, the empirical validation is not provided in this work. We plan to explore this in our future work.

## D   VISUALIZATION

**Open-vocabulary Object Detection.** We present qualitative results for open-vocabulary object detection on the OV-COCO benchmark in Fig. A2. The red boxes indicate novel categories, while the blue boxes represent base categories. These visualizations offer insights into the model's performance and its ability to detect novel objects.

**Open-Vocabulary Image Segmentation.** We present visualizations of open-vocabulary semantic segmentation results in Fig. A3. The segmentation model is trained on COCO Stuff and evaluated on the ADE20K dataset (Zhou et al., 2017).

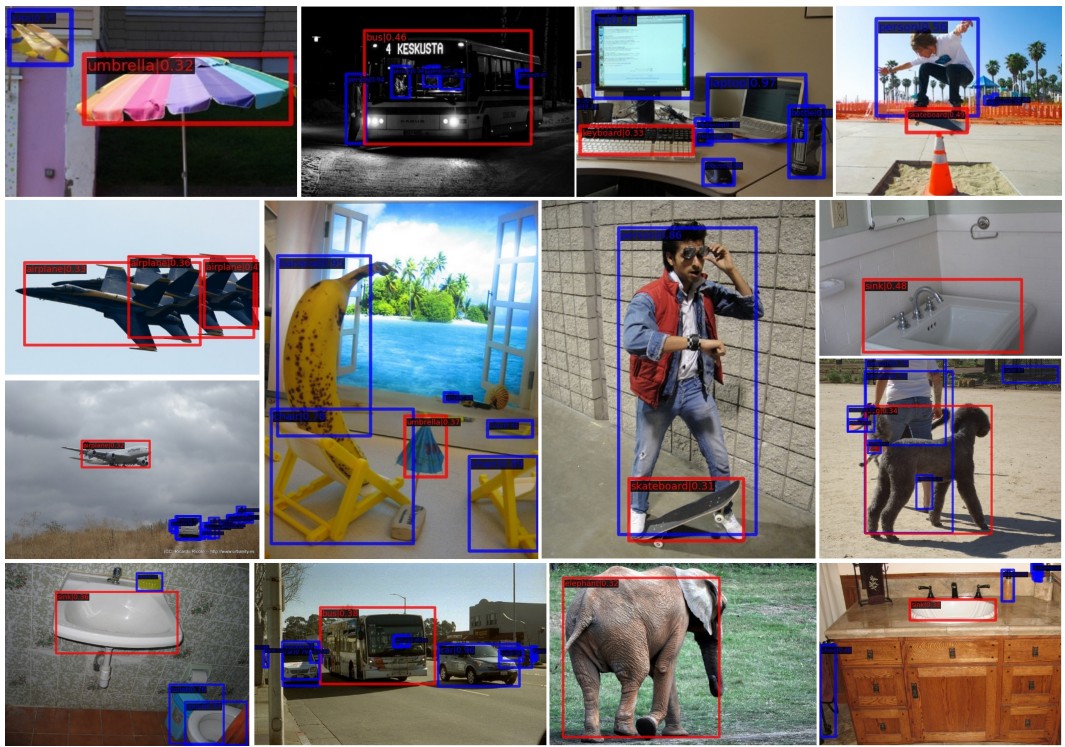

Figure A2: Visualization of object detection results. The red boxes are for the novel categories and the blue boxes are for the base categories.

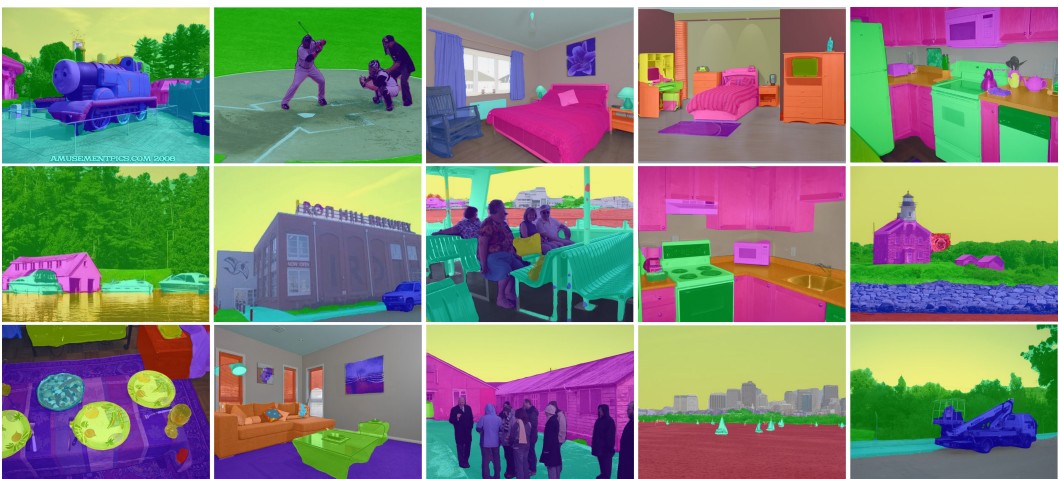

Figure A3: Visualization of image segmentation. The images are from ADE20k (Zhou et al., 2017).

