# OpenReview forum: "CLIPSelf: Vision Transformer Distills Itself for Open-Vocabulary Dense Prediction"
_ICLR.cc/2024/Conference — ICLR 2024 spotlight_

### Official Review · Reviewer_xSsC · 2023-10-30

**Soundness:** 3 good
**Presentation:** 3 good
**Contribution:** 2 fair
**Rating:** 6
**Confidence:** 4

**Summary:**

This paper proposes CLIPSelf, which is a method to adapt ViT based CLIP model trained on full images to dense prediction tasks. CLIPSelf distill a student ViT model that has good region representation from its dense feature map from a teacher ViT model that only has good full image features. Specially, CLIPSelf aligns the dense feature maps with the teacher model at crops of an image.

The proposed methods show good performance in standard benchmarks for open-vocabulary object detection, semantic segmentation and panoptic segment. It is reported to be stronger in the main metrics of the respective benchmarks to be stronger than many recent works, under the same or similar settings.

One interesting advantage of this work is that it does not need region-text pairs for training per se, unlike some recent methods such as RegionCLIP. Nevertheless, combining CLIPSelf (as pretraining step) with RegionCILP (region-text pairs for finetuning) results in further improvements for open-vocabulary detection on OV-COCO.

**Strengths:**

The main strengths of the paper are

* The paper is clearly written. It has sufficient details in the main text to reproduce the method.
* The motivation of the paper is clearly presented. The paper includes an interesting analysis on the zero-shot image classification capability using image feature and region features, using CLIP ViT and CLIP CNN, respectively. The analysis shows that CLIP ViT has much better image features for zero-shot classification, but much worse dense features for the same task. This suggests that CLIP ViT has the potential to be much stronger than its CNN counterpart, but requires additional work to adapt the full image features to dense features.
* The proposed CLIPSelf is fairly simple yet seems to be quite effective in improving the dense feature of CLIP models. This is evident in the various results on standard benchmarks from Table 3-7. The proposed CLIPSelf method seems to be among the strongest methods in standard settings in the listed benchmarks, and can improve existing methods when the CLIP pre-trained with the CLIPSelf target replaces vanilla CLIP models.
* The details in the design of CLIPSelf have been tested carefully by ablation studies. For example, the design to use randomly sampled patches in distillation is validated by the results of classifying stuff masks.

**Weaknesses:**

The main weaknesses of the paper are

* This paper does not include any studies on windowed attention based methods, such as Swin V2. This seems to have weakened the case for CLIPSelf, due to following reasons.

  * In a sense, the analysis that compares CILP ViT and CLIP CNN can also be interpreted as "CNN is more suited for dense prediction tasks". Recent empirical studies do overwhelmingly show that transformer based model outperforms CNN based models in many computer vision tasks, including dense prediction tasks. From the analysis, it appears that CILPSelf is most helpful with backbones that does not have translation equivariance built-in. But many (if not most) best models for dense predictions tasks still use backbones that have this useful properties for dense predictions (e.g. Swin V2). In fact, that is the choice for some of the best recent open-vocabulary detectors, such as GLIP (v2) and grounding DINO. The choice to only study ViT backbones leave two obvious questions open: (1) Can CLIPSelf similarly benefit Swin-based backbones? (2) How does the current method compare to SOTA methods based on Swin v2, such as GLIP and grounding DINO?
  * The square focus on ViT backbones also makes it harder, in certain cases presented in the experiments section, to interpret the comparison with prior methods. For example, in Table 3, CLIPSelf uses either ViT-B/16 or ViT-L/14 backbones. However, most of the cited methods are either from CNN backbone or Swin V2. In a few instances where the prior methods are based on ViT, the particular ViT variant is different from the two with CILPSelf.

**Questions:**

- I would like to see comparisons with GLIP, GLIP v2 and Grounding DINO.
- I would like to see if the proposed method can only improve on ViT based backbones.

---

> ### Author Response · Authors · 2023-11-16
> **R4-(1,2)**
>
> Thanks for the valuable feedback. First, please kindly refer to the discussion with **Reviewer yRsy**, who we believe have quite similar concerns and have made constructive suggestions to resolve them.
>
> **R4-1: Swin-based Backbones** Directly verifying CLIPSelf on Swin backbones is impractical as there is no published VLM based on a Swin backbone, which is agreed by **Reviewer yRsy**. Therefore, we opt for the alternative proposed by **Reviewer yRsy** to replace the global attention in CLIP ViTs with local window attention. The results are as follows.
>
> |     |        Method        |  Model   |   Boxes   | Thing Masks | Stuff Masks |  OV-COCO   |
> |:---:|:--------------------:|:--------:|:---------:|:-----------:|:-----------:|:----------:|
> |  1  | GlobalAttn(original) | ViT-B-16 | 18.2 |  20.6  |  18.4  | 17.5  |
> |  2  |      WindowAttn      | ViT-B-16 | 34.7 |  40.6  |  30.9  | 19.4  |
> |  3  | GlobalAttn+CLIPSelf  | ViT-B-16 | 72.1 |  74.4  |  46.8  | 33.6  |
> |  4  | WindowAttn+CLIPSelf  | ViT-B-16 | 73.3 |  74.9  |  48.6  | 33.6  |
>
> For a CLIP ViT enhanced with window attention, the performances on the zero-shot box/mask recognition
> and open-vocabulary detection are slightly improved (#1 and #2) but still lag behind CLIPSelf by a large margin (#2 and #3). Moreover, we show that CLIPSelf also applies to the window attention-based CLIP. Therefore, CLIPSelf is also applicable to ViT models equipped with local window attentions.
>
>
> **R4-2: Comparisons with GLIP, GLIP v2, and Grounding DINO** Although we appreciate the reviewer's suggestion on comparison with GLIP-like methods, we clarify that such comparison is infeasible and would be under an unfair setting. CLIPSelf follows the strict definition of open-vocabulary object detection that does not incorporate external region-level annotations. The only box annotations are the boxes of base categories defined in the COCO or LVIS dataset. Such a setting is widely adopted by mainstream open-vocabulary approaches [a,b,c,d,e,f,g,h, i]. In contrast, the so-called open-vocabulary object detection in GLIP, GLIP v2, and Grounding DINO is under a much looser definition, which allows training on the large detection dataset Objects365 and various grounding datasets such as Flickr30K, VG Caption, GQA, and RefCOCO. Therefore, CLIPSelf and GLIP-like methods belong to different research tracks and the comparison would not make much sense. We list some results here for reference but we do not plan to add them to the final version of our paper.
>
>
> |     |         Method         |  Model   | Box AP on LVIS |
> |:---:|:----------------------:|:--------:|:--------------:|
> |  1  |         GLIP-L         |  Swin-L  |      26.9      |
> |  2  |    Grounding-DINO-L    |  Swin-L  |      33.9      |
> |  3  |        CLIPSelf        | ViT-L/14 |      37.3      |
>
> For more details on the setting of open-vocabulary object detection, please refer to the very first paper in this direction OV-RCNN [a].
>
>
> [a] Open-Vocabulary Object Detection Using Captions, Zareian et.al., CVPR 2021.
>
> [b] Open-vocabulary Object Detection via Vision and Language Knowledge Distillation, Gu et.al., ICLR 2022
>
> [c] Open-Vocabulary DETR with Conditional Matching, Zang et.al., ECCV 2022.
>
> [d] Detecting Twenty-thousand Classes using Image-level Supervision, Zhou et.al., ECCV 2022
>
> [e] Simple Open-Vocabulary Object Detection with Vision Transformers, Minderer et.al., ECCV 2022
>
>
> [f] F-VLM: Open-Vocabulary Object Detection upon Frozen Vision and Language Models, Kuo et.al., ICLR 2023
>
> [g] Aligning Bag of Regions for Open-Vocabulary Object Detection, Wu et.al., CVPR 2023
>
>
> [h] Region-Aware Pretraining for Open-Vocabulary Object Detection with Vision Transformers, Kim et.al., CVPR 2023
>
> [i] Contrastive Feature Masking Open-Vocabulary Vision Transformer, Kim et.al., ICCV 2023

---

> ### Author Response · Authors · 2023-11-16
> **R4-(3,4)**
>
> **R4-3: Comparison With Prior Methods** We have made the benchmark comparison as fair as possible by limiting the data of distillation to only COCO and LVIS datasets. The compared ViTs (in RO-ViT) are of different architectures because they include special designs for object detection. We outperform them with Valina ViT architectures, which reflects the superiority of CLIPSelf. And please kindly note Table 3(b), where our ViT-L/14 model outperforms the larger ViT-H/16 model of RO-ViT. This is also acknowledged by reviewer **Reviewer CnGz**, who commented "More notably, CLIPSelf outperforms the recent work of RO-ViT which proposes training a new backbone catered to downstream task from scratch. I find this result particularly appealing as it shows that CLIPSelf can lightly adapt a wide range of publicly available checkpoints that were trained with relatively less consideration for the particular task of open-vocabulary object detection".
>
> To make the benchmark comparison more relevant to recent ViT-based approaches, we include the following ViT-based methods in our revised version, which are highlighted in the modified Table 3. "L" stands for the same model size (24 layers of attention modules) so the following comparison is built on the same / similar backbone compacity.
>
>
> |     |  Method  |  Model   | OV-COCO  | OV-LVIS  |
> |:---:|:--------:|:--------:|:--------:|:--------:|
> |  1  |  OW-ViT [e] | ViT-L/14 |    -     |   25.6   |
> |  2  |  RO-ViT [h] | ViT-L/16 |   33.0   |   32.1   |
> |  2  | CFM-ViT [i] | ViT-L/16 |   34.1   |   33.9   |
> |  3 | CLIPSelf | ViT-L/14 | **44.3** | **34.9** |
>
>
> [e] Simple Open-Vocabulary Object Detection with Vision Transformers, Minderer et.al., ECCV 2022
>
> [h] Region-Aware Pretraining for Open-Vocabulary Object Detection with Vision Transformers, Kim et.al., CVPR 2023
>
> [i] Contrastive Feature Masking Open-Vocabulary Vision Transformer, Kim et.al., ICCV 2023
>
>
> **R4-4: If the proposed method can only improve on ViT-based backbones** Conceptually, the proposed method applies to any VLM that has inferior region-language alignment. As the published models are mainly CNNs and valina ViTs, we only analyze these two variants in the paper as shown in Figure 1 and Figure A1. CNN-based CLIP models have shown impressive region recognition ability, so we only apply CLIPSelf to ViTs. In addition, CLIPSelf also applies to the CLIP model enhanced with local window attention which is supplemented in the rebuttal. Moreover, it is noteworthy that a valina transformer serves as a general architecture that unifies both vision and language processing without task-specific designs. Recently, there has been a trend of applying such frozen ViT-based VLMs to downstream tasks like multi-modal LLMs. In the context of adopting the freezing strategy to preserve VLM's generalization ability, CLIPSelf exhibits its advantage in preserving zero-shot ability and requiring only image data.

---

> ### Comment · Reviewer_xSsC · 2023-11-20
>
> My original comments about comparison with Swin-based backbones and GLIP/GLIP v2/ground DINO essentially seek to clarify the following question
>
> *Does CLIP-Self help only because ViT backbones lack inductive bias?*
>
> IMHO, the original submission was inadequate in addressing this important question. There were two somewhat related observations concerning this point.
>
> 1.  That its analysis seems to suggest that CNNs are naturally better at dense prediction tasks. Yet it does not seek to compare with window-attention based methods such as Swin which benefit from a design that has some translation equivariance built-in, similar to CNNs.
> 2.  That closely related works, such as GLIP/GLIPv2 and Grounding DINO, all based on Swin-V2 backbones, are not sufficiently discussed/compared with, raising the questions on whether CLIP Self only helps ViT based architectures.
>
> The additional information provided by the authors do partially address those concerns and make the paper stronger. I have some additional comments.
>
> For point 1, it is clear from the additional information that CLIP Self can improve upon some architectures with window-attention, which suggests CLIP Self is a more promising approach than was described in the original submission. This is one of the most important point to clarify, as also mentioned by reviewer yRsy. IMHO, CLIP Self would be far less interesting if it only addresses the lack of inductive bias in ViT backbones, as alternatives are available and applied to similar tasks. The comparison however still does not include Swin which would have provided more definite conclusions. I understand that this was much harder due to lack of comparable pre-trained models, but authors should clarify the reasons behind this apparent omission in the revision, so that readers won't be left guessing.
>
> For point 2, I understand and agree with the authors that the settings are different, an a direct comparison is not fair. However, would it be fair to say that in theory, assuming the conclusion in point 1 can be extrapolated to suggest that CLIP Self can improve on a wide range of architectures including swin transformers, it would be feasible to use CLIP Self to improve upon GLIP/GLIP v2 and grounding DINO, assuming sufficient time and resources invested? If so, the lack of comparison is a limitation of this work. The relation between CLIP Self and those methods should be clearly discussed in the paper. This is particularly relevant, again, as I feel CLIP Self would be less interesting if it is only addressing inductive bias of ViT, so the relation to other window-attention based approaches addressing similar (although different) settings should be acknowledged and analyzed whenever applicable.

---

> ### Author Response · Authors · 2023-11-21
> **Response to the additional comments**
>
> Thanks for the timely response! We greatly appreciate your kind clarification on the concerns and the additional comments provided, which we find vitally important to establish the soundness of this paper. We have revised the PDF accordingly.
>
> **Point 1**
>
> Thanks for acknowledging the significance of our additional experiments on **applying CLIPSelf to ViTs with window attention**. We have now moved these results and discussions to the main text of our latest revision. Besides, we have also followed your kind suggestion on clarifying the reasons why a direct comparison with Swin-based VLMs cannot be provided to avoid confusion. These contents are now in Sec 4.4 in the latest revised PDF.
>
> **Point 2**
>
> (1) we thank the reviewer for understanding that a direct benchmark comparison with GLIP-like methods cannot be established due to different experimental settings.
>
> (2) We have provided the alternative that develops window attention based ViTs from public CLIP models, which has helped verify CLIPSelf's promising applicability.
>
> (3) Given the addtional experiments on window attention based architectures, we believe the extrapolation that "CLIP Self can improve on a wide range of architectures including swin transformers" holds.
>
> We envision that CLIPSelf has good potential to be applied to GLIP-like methods. The self-distillation can be a strong supplement or replacement of the pseudo labelling processes in these methods. However, a further study on the architeture of GLIP and Grouding DINO as well as on the vision and language representations in these models would be a pre-requisite of applying the self-distillation. Because these works only treat Swin as a visual backbone and learn region-language grounding in a more sophisticated cross-modality decoder. And there should also be study on the resources needed for self-distllation and on the specific stage (pre-training or fine-tuning) the self-distillation is applied to.
>
> These studies would be out of the scope of this paper given limited time and resources. However, in the revision (Sec.C), we have clearly pointed out this as the limitation of our work, which we plan to explore in the future.
>
> (4) Following the reviewer's insightful suggestion on clearly discussing CLIPSelf's "relation to other window-attention based approaches addressing similar settings whenever applicable",  we have provided a comprehensive discussion on these relevant works including the Swin-based GLIP and Grounding DINO. It is noteworthy that we contextualize this discussion under "**Vision Transformers in Open-Vocabulary Learning**", which is a new paragraph in the related work section (Sec.2). We discuss a series of ViT-related works in the context of open-vocabulary dense prediction, and analyze their progress and limitations compared with CLIPSelf. The reviewer might be interested in this revised part of related work.

---

> ### Author Response · Authors · 2023-11-23
>
> Dear Reviewer xSsC,
>
> The discussion phase will end very soon. we would appreciate a final response from you for the discussion and we are looking forward to a justification of your final rating for this paper.
>
> Best regards,
> Authors of Submission 5422

---

> > ### Comment · Reviewer_xSsC · 2023-11-23
> >
> > I appreciate the further clarification and the candor of the authors in the response.
> >
> > The inclusion of window attention experiments hint at this work being broadly applicable, but it remains an open question on whether it can be applied directly to practical architectures that have better inductive biases for dense predictions, such as Swin transformers. However, as pointed out by the authors and other reviewers, such direct experiments are difficult, and the empirical evidences provided in the submission itself are against strong baselines in the same settings. Hence it is not unreasonable to expect a follow-up work to address those remaining questions.
> >
> > Given this, I think it is okay to accept this paper in its latest revisions. I will change my official rating accordingly.

---

> > > ### Author Response · Authors · 2023-11-23
> > >
> > > Thanks for the suggestions in the discussion phase! We'll keep on exploring the remaining issues in future works!

---

### Official Review · Reviewer_CnGz · 2023-11-01

**Soundness:** 3 good
**Presentation:** 4 excellent
**Contribution:** 3 good
**Rating:** 8
**Confidence:** 5

**Summary:**

This paper presents a simple approach to improve the transferability of CLIP ViT models to open-vocabulary object detection and image segmentation.
While CLIP ViT shows strong performance in image classification (as compared to CLIP ConvNets),
its region-level representation is known to underperform in dense prediction tasks.
The authors first demonstrate this shortcoming through a simple experiment,
then propose a self-distillation approach named CLIPSelf to remedy this.
In their experiments, the authors apply the CLIPSelf procedure to fine-tune the CLIP ViT checkpoints made public by OpenAI,
and then use the fine-tuned weights for open-vocabulary object detection and image segmentation.
Empirical results show that visual representations from CLIPSelf are better than CLIP for dense prediction tasks,
and CLIP ViTs fine-tuned by CLIPSelf procedure are highly competitive on open-vocabulary detection benchmarks like COCO and LVIS.

**Strengths:**

**Note:**
I reviewed this paper for the NeurIPS 2023 conference.
I will contextualize my current review based on the changes the authors have made since their previous submission.
The main idea of CLIPSelf has been unchanged since NeurIPS 2023 submission --
the authors have significantly revamped the experimental study with more suitable benchmarks and baselines,
and added more implementation details in the draft.

I identified some key strengths of the proposed approach in the NeurIPS 2023 submission, they still apply to the current submission:

- The proposed approach (CLIPSelf) is conceptually simple and shows large improvements over CLIP on multiple evaluations shown in the paper.
- CLIPSelf does not require any form of annotations beyond just images, this relaxation simplifies the data and modeling pipeline.
- Authors have performed comprehensive evaluations with multiple downstream tasks involving dense predictions to show the versatility of the proposed approach.

I find notable other strengths in the current submission:

- Empirical results on open-vocabulary detection are quite strong. CLIPSelf outperforms prior works that use equivalent frozen backbones (e.g. F-VLM, CORA).
- More notably, CLIPSelf outperforms the recent work of RO-ViT which proposes training a new backbone catered to downstream task _from scratch_.
  I find this result particularly appealing as it shows that CLIPSelf can lightly adapt a wide range of publicly available checkpoints that were trained with relatively less consideration for the particular task of open-vocabulary object detection.
- The writing clarity has also improved in this submission.

**Weaknesses:**

My review mainly contrasts the previous submission (NeurIPS 2023) with the current ICLR 2024 submission --
since the main story of the paper is mostly unchanged, I do not find any significant outstanding concerns.

I have a few outstanding concerns and suggestions, which may further improve the paper.
I am curious to hear the authors' thoughts on these and am willing to engage in the discussion:

1. **Is self-distillation on task dataset (COCO) necessary?**
Main CLIPSelf experiments use images from COCO/LVIS dataset to perform self-distillation.
The authors should consider disentangling the impact of self-distillation in isolation,
as compared to self-distillation using images from the target downstream task.
If CLIPSelf does not _require_ COCO images, then it strengthens the contribution --
the authors may call for future works to "fine-tune" their CLIP ViT for an additional epoch to improve their out-of-the-box usability for dense prediction tasks.
It would be even better if CLIP ViTs retain their original zero-shot classification and retrieval performance after such self-distillation.
If the authors wish to do this experiment in the rebuttal, they may consider a large dataset of diverse unlabeled images
(e.g. images from datasets with approx. 10-20 million samples like Conceptual Captions, RedCaps, YFCC-15M, Segment Anything dataset, etc.)

2. **Accounting for the 'padding' patches during distillation.**
Current state-of-the-art object detectors using plain ViTs (e.g. ViTDet and SAM) typically pad input images to a square.
Any non-square input images are made square by adding zero pixels to the right and bottom after color normalization.
Pre-trained CLIP ViTs have not seen such patches so they encounter a train-test mismatch if they are kept frozen in the object detector.
Have the authors considered randomly adding padding regions to image patches while performing self-distillation?
I believe that accounting for this detail will ease the adaptation of CLIP ViTs for open-vocabulary object detection.

**Summary:**
The authors have diligently incorporated the reviewer feedback from the previous conference submission along with other significant changes.
In my opinion, these changes have strengthened the technical contributions and improved the presentation of this paper.
I have a few other outstanding concerns but the technical strengths already outweigh them.
I believe this paper is ready to be a part of the published literature -- I am leaning towards an acceptance.

**Questions:**

Minor questions and suggestions:

- Figure 3 (b) non-standard word use: "initiate" -> "initialize"
- Figure A2 caption is incorrect: "image segmentation" -> "object detection"

**Details Of Ethics Concerns:**

The broader impact statement is short and concise, but I believe it is sufficient.

---

> ### Author Response · Authors · 2023-11-16
>
> We first convey our sincere gratitude for your services in both conferences and valuable feedback during the two review processes.
>
> **R3-1: Distillation on Non-COCO Dataset** We do the distillation only on the COCO/LVIS dataset mainly to ensure a fair comparison with existing methods, especially those distillation-based and frozen CLIP-based approaches. Here we add the experiment on the CC3M dataset, which includes 28 million images. The results are as follows.
>
>
> |     | Distillation Dataset |        Model        | Boxes  | Thing Masks | Stuff Masks | OV-COCO | OV-LVIS |
> |:---:|:--------------------:|:-------------------:|:------:|:-----------:|:-----------:|:-------:|:-------:|
> |  1  |          -           | ViT-B-16 (EVA-CLIP) |  18.2  |    20.6     |    18.4     |  17.5   |  11.5   |
> |  2  |         COCO         | ViT-B-16 (CLIPSelf) |  72.1  |    74.4     |    46.8     |  33.6   |  25.3   |
> |  3  |         CC3M         | ViT-B-16 (CLIPSelf) |  72.1  |    74.5     |    49.5     |  35.8   |  26.6   |
>
> As shown in the table, CLIPSelf exhibits consistent improvement on both zero-shot box and mask recognition as well as open-vocabulary detection on the COCO/LVIS dataset.
>
> **R3-2: 'Padding' Patches** Thanks for the suggestion. We have already done this in both self-distillation and detection training by adding zero pixels to the right and bottom of non-square images after color normalization. We'll make this clear in the implementation details.
>
> **R3-3: Typos** Thanks for pointing out the typos in Figure 3 and Figure A2. We'll correct them in the final version.

---

> ### Author Response · Authors · 2023-11-23
>
> Dear Reviewer CnGz,
>
> The discussion phase will end very soon. we would appreciate it if you could have a look at the discussion and the revisions of this paper before the deadline.
>
> Best regards, Authors of Submission 5422

---

> > ### Comment · Reviewer_CnGz · 2023-11-23
> > **Thank you for the rebuttal!**
> >
> > I thank the authors for conducting the additional experiments and incorporating all reviewer suggestions in their paper, which have strengthened the paper. I will maintain my original score.

---

> > > ### Author Response · Authors · 2023-11-23
> > >
> > > Thanks for the suggestions in the discussion phase! We'll keep on exploring the remaining issues in future works!

---

### Official Review · Reviewer_BRE4 · 2023-11-02

**Soundness:** 3 good
**Presentation:** 3 good
**Contribution:** 3 good
**Rating:** 6
**Confidence:** 4

**Summary:**

In this work, the authors proposed a simple yet effective method for learning fine-grained visual-language representations for open-vocabulary detection and segmentation tasks. Given that the original CLIP-based VIT models have a great degradation of performance for zero-shot fine-grained image understanding, the authors explored a cheap way of refining the visual representation from the original CLIP vision encoder by distilling the image-patch representation to regional dense feature representations. Through such a simple strategy, the region-level visual-semantic representation ability is immediately unleashed. According to the extensive evaluations of different tasks and settings, the refined representations significantly outperform the original ones and other state-of-the-art methods. Overall, I think the proposed method called CLIPSelf is a very simple yet effective way to refine the existing models, which could be a good way of learning better representations for open-vocabulary tasks under low budgets.

**Strengths:**

1. The proposed method CLIPSelf, distilling CLIP models for fine-grained representation is simple yet effective. It is motivated by an interesting observation in the study of CLIP representation for image-level tasks to region-level tasks. The author also observed that cropping image patches significantly outperform feature pooling for ViT vision encoders. As a result, the authors proposed to distill the fine-grained understanding of visual content in image patches to the regional representations in the hidden feature space.

2. The authors conducted extensive experiments to validate the effectiveness of the proposed methods, spanning from open-vocabulary object detection to semantic segmentation and then panoptic segmentation. It shows that the refined visual-semantic representations by CLIPSelf bring significant gain over the baseline model and also outperform strong previous works.

**Weaknesses:**

1. There are some related works that already explored the way of distilling patch-level representations to regional feature-level representations, such as ZeroSeg proposed in "Exploring Open-Vocabulary Semantic Segmentation without Human Labels. Chen et al. ICCV 2023". The overall methodology proposed in this work resembles the one in ZeroSeg, in that both attempted to distill the CLIP knowledge from patch to feature map and for open-vocabulary segmentation tasks.

2. It turns out that the authors only used the COCO training set for distillation. A doubt on this setting is that it is probably very favorable to downstream tasks like COCO detection and segmentation, and ADE20K as well. Though this in-domain distillation brings a significant boost, the authors should have more studies and evaluations on more downstream datasets from various domains and also try using non-COCO training sets for distillation, which seems to be more realistic.

**Questions:**

Overall,I have two main questions:

1. How to distinguish this work from ZeroSeg regarding the core learning algorithm?

2. Are the improvements over CLIP baselines sourced from in-domain training? What if we use another dataset like CC3M for the distillation?

One minor question:

How to adapt the VIT models to different input image resolutions?

---

> ### Author Response · Authors · 2023-11-16
>
> **R2-1: Discussion on ZeroSeg** Although ZeroSeg and CLIPSelf are similar in that both approaches perform distillation based on image patches instead of human annotations, we would like to categorize ZeroSeg to distillation-based open-vocabulary approaches that typically transfer the knowledge of CLIP models to downstream object detection models or image segmentation models, including ViLD[a], and the follow-ups like OV-DETR[b] and BARON[c]. Therefore, such approaches are quite task-specific as the students vary in tasks and architectures. In contrast, CLIPSelf is generally applicable to various downstream tasks with the teacher and student being the exactly same model. And the zero-shot recognition ability is better-preserved thanks to this self-distillation design, which is verified by the benchmark results. Abstractly, CLIPSelf is at a point between upstream vision-language pretraining and downstream detection/segmentation applications like ZeroSeg and ViLd.
>
> In addition, there are also clear differences in the technical details. As the segment tokens in ZeroSeg do not preserve explicit location information, ZeroSeg applies both a global-local distillation loss and a segment-matching loss. In comparison, the design of CLIPSelf is much simpler due to the direct correspondence between image crops and locations on the dense feature map, with a single loss that simply maximizes cosine similarities between dense features and the corresponding crop features. Moreover, the distillation of CLIPSelf would be more precise as we do not require the noisy matching process that is necessary for ZeroSeg.
>
> [a] Open-vocabulary Object Detection via Vision and Language Knowledge Distillation, Gu et.al., ICLR 2022
>
> [b] Open-Vocabulary DETR with Conditional Matching, Zang et.al., ECCV 2022.
>
> [c] Aligning Bag of Regions for Open-Vocabulary Object Detection, Wu et.al., CVPR 2023
>
> **R2-2: Non-COCO datasets** Thanks for the suggestion. We do the distillation only on the COCO/LVIS dataset mainly to ensure a fair comparison with existing methods, especially those distillation-based and frozen CLIP-based approaches. For various downstream datasets, we already include such results in Table 4,5,6, where the LVIS-trained or COCO-trained models are evaluated on other datasets. For a non-COCO dataset in self-distillation, we supplement the experiment using the CC3M dataset and the results are as follows.
>
>
> |     | Distillation Dataset |        Model        | Boxes  | Thing Masks | Stuff Masks | OV-COCO | OV-LVIS |
> |:---:|:--------------------:|:-------------------:|:------:|:-----------:|:-----------:|:-------:|:-------:|
> |  1  |          -           | ViT-B-16 (EVA-CLIP) |  18.2  |    20.6     |    18.4     |  17.5   |  11.5   |
> |  2  |         COCO         | ViT-B-16 (CLIPSelf) |  72.1  |    74.4     |    46.8     |  33.6   |  25.3   |
> |  3  |         CC3M         | ViT-B-16 (CLIPSelf) |  72.1  |    74.5     |    49.5     |  35.8   |  26.6   |
>
> As shown in the table, CLIPSelf exhibits consistent improvement on both zero-shot box and mask recognition as well as open-vocabulary detection on the COCO/LVIS dataset. We're glad that this experiment is also advised by **Reviewer CnGz**, who suggested that the experiment "strengthens the contribution -- the authors may call for future works to 'fine-tune' their CLIP ViT for an additional epoch to improve their out-of-the-box usability for dense prediction tasks".
>
> **R2-3: Different Resolutions** The attention modules are agnostic to input size, we only need to upsample the positional embeddings to fit different image resolutions.

---

> ### Author Response · Authors · 2023-11-21
> **Discussion on ZeroSeg**
>
> We have added the discussion on ZeroSeg to the related work (Sec.2) in the latest revision. And the experiment on CC3M is also added to the main text of the paper in Sec 4.4.

---

> ### Author Response · Authors · 2023-11-23
>
> Dear Reviewer BRE4,
>
> The discussion will end very soon. we would appreciate a final response from you for the discussion and we are looking forward to a justification of your final rating for this paper.
>
> Best regards,
> Authors of Submission 5422

---

### Official Review · Reviewer_yRsy · 2023-11-09

**Soundness:** 3 good
**Presentation:** 4 excellent
**Contribution:** 3 good
**Rating:** 8
**Confidence:** 4

**Summary:**

The paper proposed to address the problem of adapting vision Transformers pre-trained with image-language alignment (e.g., CLIP) for dense prediction tasks (e.g., object detection, semantic segmentation). The key idea is to distill features from the pre-trained Transformer model itself, in which region representation from a dense feature map matches those from cropped regions (using the pre-trained model). The proposed method was evaluated on multiple open-vocabulary object detection and semantic segmentation benchmarks. The results are solid.

**Strengths:**

* The paper addressed an important and trendy topic on adapting large-scale pre-trained vision-language models (i.e., foundation models).

* The paper is well written. Key concepts are clearly described. Technical details are easy to follow.

* The core idea is well motivated and quite intuitive.

* The experiments are extensive (three open-vocabulary dense prediction tasks across multiple datasets + ablations). The results are impressive.

**Weaknesses:**

It is somewhat unclear why the proposed method, which is conceptually simple, can work well. My bet is that the proposed self-distillation encourages localized visual features, i.e., features on the dense map capture local region representations (instead of global image representations). This is partially backed up by Fig 1 (b). The gap between dense features and cropped image features is negligible for CNN (where the features are localized), yet rather significant for ViT. Part of the reason lies in the design of ViT using global attention, where the dense features are diffused in a way that each spot encodes a global image representation. There might be ways to validate this explanation. For example, one possibility is a retrieval experiment using features on the dense maps. Features on dense maps from pre-trained CLIP ViT might yield similar images instead of similar regions, while features after distillation may better retrieve regions. It will be great if the authors can comment on this reasoning here and discuss the key intuition in the paper.

**Questions:**

If the explanation described in the weaknesses section is true or partially true, it brings two additional questions.

* First, do those new localized representations (after distillation) still preserve the zero-shot recognition capacity of CLIP? One interesting experiment is to evaluate those features directly for zero-shot object detection / segmentation (e.g., see Table 4 in RegionCLIP).

* Second, is the proposed method a remedy to a particular design (ViT with global self-attention)? This question is perhaps more fundamental. What if a CLIP model is trained using ViT with local self-attention (e.g., using Swin Transformers)? Will they still benefit from the method? This is much harder to validate directly, as there is no pre-trained models available. Yet one possible alternative is to modify the pre-trained ViT by replacing global attention with local ones, and then train for those open-vocabulary tasks. (Indeed, MaskCLIP can be considered as a special case of this modification, where the global attention at the end is replaced by a local attention of size 1x1.) It will be interesting to see how the proposed method compares to this alternative.

---

> ### Author Response · Authors · 2023-11-16
> **Experiments**
>
> **R1-1: Retrieval Experiment** We greatly appreciate the reviewer's valuable insight that "the dense features are diffused in a way that each spot encodes a global image representation". And we conduct the retrieval experiment proposed by the reviewer.
>
> Specifically, we first encode images and regions annotated in COCO's validation split. Then we extract dense feature maps of the images and let each location in the feature map retrieve the image and region that it belongs to by calculating cosine similarity between the location feature and the encoded image or region features. For each retrieval, we provide 50 samples (1 positive and 49 negative samples) and calculate recall at 1, 5, and 10, respectively. The results are as follows.
>
>
> |        Model        | Image R@1 | Image R@5 | Image R@10 | Region R@1 | Region R@5 | Region R@10 |
> |:-------------------:|:---------:|:---------:|:----------:|:----------:|:----------:|:-----------:|
> | ViT-B-16 (EVA-CLIP) |   67.0    |   85.6    |    91.4    |    34.7    |    60.5    |    71.4     |
> | ViT-B-16 (CLIPSelf) |   28.4    |   50.3    |    62.5    |    57.1    |    80.1    |    85.8     |
>
> The experimental results are a direct verification of the reviewer's hypothesis. For the original CLIP model, the dense features are well-matched with the corresponding images, indicating that "each spot encodes a global image representation". The model refined by CLIPSelf yields dense features that match the corresponding regions instead of images, revealing the transfer of knowledge from global image representation to local region representation. We add this experiment to the paper to better interpret our intuition.
>
>
> **R1-2: Zero-shot Recognition** Although we do not report the APs of zero-shot object detection like Table 4 in RegionCLIP, we have similar experimental results that also measure the zero-shot recognition ability in Table 2 and Table 6. Table 2 shows the accuracy of classifying GT boxes and masks using dense features of CLIP, which is quite similar to the results of using GT proposals in Table 4 of RegionCLIP, except that we use mAcc instead of APs. Table 6 includes results that only use CLIP scores during inference, meaning that the mask proposals predicted by ODISE are classified by only using CLIP's dense features. These results in Table 6 are exactly the zero-shot segmentation mentioned by the reviewer.
>
>
> In addition to the existing experimental results in the paper, we also add a zero-shot objection detection experiment under the same setting of Table 4 in RegionCLIP using an external RPN to generate region proposals. The results are as follows.
>
> |        Model        | AP50_novel |  AP50_base  |   AP50    |
> |:-------------------:|:----------:|:-----------:|:---------:|
> | ViT-B-16 (EVA-CLIP) |    12.7    |    12.3     |   12.4    |
> | ViT-B-16 (CLIPSelf) |    35.5    |    32.1     |   33.0    |
>
> The improved region recognition performance reveals that the zero-shot recognition ability is transferred from global image representation to local region representation.
>
> **R1-3: Window Attention** Thanks for the insightful suggestion, we replace the global attention in original CLIP models with local window attention, which slightly improves the recognition ability of the dense features as shown in the following table (#1 and #2). We also train open-vocabulary detection using the window attention (the rightmost column).
>
>
> |     |        Method        |  Model   |   Boxes   | Thing Masks | Stuff Masks |  OV-COCO   |
> |:---:|:--------------------:|:--------:|:---------:|:-----------:|:-----------:|:----------:|
> |  1  | GlobalAttn(original) | ViT-B-16 | 18.2  |  20.6 |  18.4 | 17.5  |
> |  2  |      WindowAttn      | ViT-B-16 | 34.7  |  40.6 |  30.9 | 19.4  |
> |  3  | GlobalAttn+CLIPSelf  | ViT-B-16 | 72.1  |  74.4 |  46.8 | 33.6  |
> |  4  | WindowAttn+CLIPSelf  | ViT-B-16 | 73.3  |  74.9 |  48.6 | 33.6  |
>
> The performance of simply replacing global attention with local window attention lags largely behind using CLIPSelf for self-distillation (#2 and #3). Moreover, we find CLIPSelf can consistently improve the CLIP model whose global attention is replaced with window attention (#2 and #4).

---

> > ### Author Response · Authors · 2023-11-16
> > **Intuition**
> >
> > **R1-4: Intuition** Based on the current paper and the supplemented experimental results, we re-iterate our motivation of remedying the incompetent dense representation of modern ViT-based vision-language models. Moreover, there has been a trend of applying frozen VLMs to downstream tasks [a,b,c,d], prominently in the hot topics like open-vocabulary detection/segmentation and multi-modal LLMs. In the context of adopting the freezing strategy to preserve VLM's generalization ability, CLIPSelf exhibits its advantage in preserving zero-shot ability and requiring only image data.
> >
> > [a] F-VLM: Open-Vocabulary Object Detection upon Frozen Vision and Language Models, Kuo et.al., ICLR 2023
> >
> > [b] Frozen CLIP Models are Efficient Video Learners, Lin et.al., ECCV 2022
> >
> > [c] Frozen CLIP Model is An Efficient Point Cloud Backbone, Huang et.al. Arxiv 2022
> >
> > [d] BLIP-2: Bootstrapping Language-Image Pre-training with Frozen Image Encoders and Large Language Models, Li et.al., Arxiv 2023

---

> > > ### Comment · Reviewer_yRsy · 2023-11-22
> > >
> > > I thank the authors for responding to my comments. The new experiments were nicely done and addressed my previous questions. The paper has a neat idea and the revision looks solid. I will maintain my previous rating and recommend to accept this paper.

---

> > > > ### Author Response · Authors · 2023-11-23
> > > >
> > > > Thanks for the suggestions in the discussion phase! We'll keep on exploring the remaining issues in future works!

---

### Author Response · Authors · 2023-11-16
**To All Reviewers**

We thank all reviewers for the valuable feedback, which would further improve the quality of this paper.

**Reviewer yRsy** suggested the retrieval experiment that provides new insights into the CLIP ViT's dense features and the experiment on local window attention to give CLIPSelf a more rigorous validation.

**Reviewer BRE4** and **Reviewer CnGz** advised performing self-distillation on a non-COCO dataset, further validating CLIPSelf's effectiveness in transferring CLIP's zero-shot recognition ability to local representations.

**Reviewer xSsC**'s concern about the comparison with prior methods motivates us to include more ViT-based approaches and reorganize the benchmark results in Table 3, which makes the comparison more relevant to recent ViT-based methods and better establishes CLIPSelf's advantage in adapting ViTs for open-vocabulary detection.

These modifications and additions are highlighted in the revised PDF using purple color. We also individually respond to each reviewer in the discussion boxes.

---

### Author Response · Authors · 2023-11-21
**Further Revision of the PDF**

Dear reviewers,

We have updated the PDF with further revision in the main text.

**1.** We moved some important supplemented experiments and discussions to the main text, including the exploration on window attention (proposed by **Reviewer yRsy**) and the experiment on CC3M (proposed by **Reviewer BRE4** and **Reviewer CnGz**). Now they are in Sec. 4.4 and highlighted in purple.

**2.** We re-organized the Sec. 2 (related work) with a new paragraph contextualizing the disccusion on relevant ViT works, which is suggested by **Reviewer xSsC** and also highlighted in purple.

**3.** We added the discussion on the concurrent ZeroSeg to related work (Sec.2 Open-Vocabulary Dense Prediction) as suggested by **Reviewer BRE4**, and highlighted these contents in purple.

**4.** The transfer evaluation of open-vocabulary detector was moved to appendix to fit the 9-page limitation as we observe less attention to this part from the reviewers.

As the deadline approaches, we cordially invite all reviewers to engage in the dicussion. Again, we sincerely thank all the feedback that has made this paper better.

---

### Author Response · Authors · 2023-11-23

Dear all,

We understand that some of you may be preparing for the CVPR submission's supplemental material or planning for the thanksgiving holiday. But we would appreciate it if you could take a few minutes to give a final summary of the discussion phase regarding the revisions we made, and have a justification of your final ratings of this paper.

Wish you all a good thanksgiving day!

Best regards,
Authors of Submission 5422

---

### Meta-Review · Area_Chair_5xMe · 2023-12-02

**Metareview:**

The paper proposed a simple distillation method (CLIPSelf) for Clip ViT model. CLIPSelf empowers ViTs to distill itself by aligning a region representation extracted from its dense feature map with the image-level representation of the corresponding image crop. The distilled ViT achieves new state-of-the-art performance on open-vocabulary object detection, semantic segmentation, and panoptic segmentation across various benchmarks.
Pros:
* The proposed method is simple and very effective.
* Very thorough experimental validation on open-vocabulary object detection, semantic segmentation, and panoptic segmentation.
* The analysis and intuition are clear.
Cons:
* Some relevant work like ZeroSeg is not discussed.

During the rebuttal, the authors discussed ZeroSeg and addressed many questions raised by the reviewers.
Hence, all reviewers give positive ratings. AC also recommends acceptance.

**Justification For Why Not Higher Score:**

The proposed method achieves impressive results.
However, this is not the first work proposing to distill patch-level representations to regional feature-level representation.

**Justification For Why Not Lower Score:**

The proposed method achieves significant performance gain and can be applied to open-vocabulary object detection, semantic segmentation, and panoptic segmentation.

---

### Decision · Program_Chairs · 2024-01-16

Accept (spotlight)